# DYNAMICS BASED NEURAL ENCODING WITH INTER-INTRA REGION CONNECTIVITY

## ABSTRACT

Extensive literature has drawn comparisons between recordings of biological neurons in the brain and deep neural networks. This comparative analysis aims to advance and interpret deep neural networks and enhance our understanding of biological neural systems. However, previous works did not consider the time aspect and how the encoding of video and dynamics in deep networks relate to the biological neural systems within a large-scale comparison. Towards this end, we propose the first large-scale study focused on comparing video understanding models with respect to the visual cortex recordings using video stimuli. The study encompasses more than two million regression fits, examining image *vs.* video understanding, convolutional *vs.* transformer-based and fully *vs.* self-supervised models. Our study resulted in both, insights to help better understand deep video understanding models and a novel neural encoding scheme to better encode biological neural systems. We provide key insights on how video understanding models predict visual cortex responses; showing video understanding better than image understanding models, convolutional models are better in the early-mid visual cortical regions than transformer based ones except for multiscale transformers and that two-stream models are better than single stream. Furthermore, we propose a novel neural encoding scheme that is built on top of the best performing video understanding models, while incorporating inter-intra region connectivity across the visual cortex. Our neural encoding leverages the encoded dynamics from video stimuli, through utilizing two-stream networks and multiscale transformers, while taking connectivity priors into consideration. Our results show that merging both intra and inter-region connectivity priors increases the encoding performance over each one of them standalone or no connectivity priors. It also shows the necessity for encoding dynamics to fully benefit from such connectivity priors.

## 1 INTRODUCTION

There has been a recent increase in studies that compare how deep neural networks process input stimuli to the processing that occurs in the brain whether in humans (Zhou et al., 2022; Conwell et al., 2021b; Schrimpf et al., 2018; Cichy et al., 2019; 2021), non-human primates, or rodents (Conwell et al., 2021a; Schrimpf et al., 2018). The benefits of these studies are two-fold. First, such a comparison can be used to interpret and have a better understanding of black-box deep neural networks and even provide inspiration on how to improve them. Second, it can provide a better understanding and encoding of biological neural systems. Towards achieving this, recent benchmarks have been released to improve the capabilities of machine learning models in neural encoding and comparing them to biological neural systems (Schrimpf et al., 2018; Cichy et al., 2019; 2021; Gifford et al., 2023). The current established benchmark from The Mini-Algonauts Project 2021 (Cichy et al., 2021), has provided neuro-imaging data for the brain responses from participants watching short video clips. A further extension of the aforementioned dataset (Lahner et al., 2024) provided exhaustive analysis with additional cortical regions for the ventral and dorsal streams. It also provided empirical evidence that temporal information was captured in fMRI recordings of the visual cortex for participants watching video stimuli, by studying the effect of frame shuffling and analyzing the first and last second of these videos. The former showed that shuffling lead to degraded performance confirming that there are dynamics captured in the recorded fMRI, while the latter showed that these fMRI recordings encode temporally distinct early and late video snapshots. These developments and

benchmarks can enable neuroscientists to study how the brain understands dynamics (e.g., motion), which is critical to study in neural encoding.

Machine learning models have been investigated in encoding such neuro-imaging data (Lahner, 2022; Zhou et al., 2022; Lahner et al., 2024). However, these studies either focused on a small-scale comparison of a few video-understanding models or conducted their study on single image-based models, neglecting the dynamics aspect. More importantly, previous studies did not conduct a systematic analysis of different model families. To address this gap, we propose the first large-scale comparative study of video understanding in neural encoding that encompasses more than two million regression fits (this refers to source models, their layers, target models, regions, and voxels count). Our study takes various properties into consideration where we study image *vs.* video understanding, convolutional *vs.* transformer based, single-stream *vs.* two-stream and fully supervised *vs.* self-supervised ones. Furthermore, we not only study these video understanding models to encode cortical regions recordings, but we also study them in a setup where the target is another artificial neural network, inspired by the single image understanding study (Han et al., 2023). Our results show that video-understanding models are better than image-understanding ones in modelling the human visual cortex recordings. Specifically, two-stream convolutional models and multiscale transformers were the best ones. Interestingly, we are the first to demonstrate the effect of multiscale processing in transformers that improves its ability to capture low-level cues (e.g. oriented gradients) consequently improving its performance in the early regions.

Finally, we devise a novel neural encoding mechanism that takes into account encoded dynamics and connectivity priors. Voxel connectivity started to get explored in neural encoding in a few recent works (Mell et al., 2021; Xiao et al., 2022). However, previous works used this approach in a limited scope in which they studied the connectivity between pairs of regions without considering the connectivity across all regions at once. They also did not consider the combined effect of the local connectivity within the same region (intra-region connectivity) and the global connectivity across the different regions (inter-region connectivity).

In summary, our contributions are threefold:

- We showcase the first large-scale study of deep video understanding models on two datasets for the human visual cortex where the models include convolutional *vs.* transformer-based, single *vs.* two stream and fully *vs.* self-supervised. Our study is comprehensive with more than 35 models from various families and more than two million regression fits, unlike the recent work (Lahner et al., 2024) that showed only three models with limited types.
- We establish an artificial neural network target environment setup using image and video understanding models as the target and a a biological target environment setup with the target model as the human visual cortex.
- We propose a novel fully integrated encoding model that takes into account intra and inter-region connectivity priors in the visual cortex with features extracted from pre-trained video understanding models. We also show that encoding dynamics is an important aspect to enable the full utilization of such connectivity priors.

## 2 RELATED WORK

**Biological neural systems encoding.** Brain encoding is concerned with mapping the input stimuli to the neural activations in the brain. Learning this mapping has been heavily investigated in the literature (Zhou et al., 2022; Conwell et al., 2021b; Lahner, 2022), where most of the approaches conduct a form of deep regression. The study of brain encoding has been advanced by the release of naturalistic neuroscience datasets and benchmarks, with text, audio, image, or video stimuli. One of the well-established benchmarks that studied how deep networks compare to biological neural systems is the Brain-Score benchmark and framework (Schrimpf et al., 2018) which relied on grayscale image stimuli. Another recent well established dataset and benchmark, The Algonauts project (Cichy et al., 2019; 2021; Gifford et al., 2023), released datasets and challenges that focused on stimuli as natural objects images (Cichy et al., 2019), action videos (Cichy et al., 2021; Lahner et al., 2024) and natural scenes (Gifford et al., 2023). In these benchmarks, fMRI responses were recorded from different subjects and used to study how the human brain encodes these different kinds of stimuli. In this work, we focus specifically on recorded fMRI data for participants watching short video clips from

Mini-Algonauts 2021 (Cichy et al., 2021). Other benchmarks were released with video stimuli focus (Popham et al., 2021; Lahner et al., 2024), including the extension of the aforementioned dataset and the most recent benchmark, BOLD moments (Lahner et al., 2024). BOLD moments provided an exhaustive analysis and an improved dataset, as such we also evaluate on it as part of our study. Recent works have also investigated the ability of deep networks to regress on the brain responses for different stimuli (Conwell et al., 2021b; Zhou et al., 2022), where one approach (Zhou et al., 2022) focused on video stimuli. However, they mainly worked with single-image deep neural network architectures. In this work, we follow this approach closely, but we focus on studying video understanding models to draw insights on how the brain understands actions and models dynamics. While some works in neuroscience studied the time aspect (Zhuang et al., 2021; Nishimoto et al., 2011; Khosla et al., 2021; Nishimoto, 2021; Lahner et al., 2024; Güçlü & Van Gerven, 2017; Shi et al., 2018; Sinz et al., 2018; Huang et al., 2023), they did not focus on large-scale comparison. Our work focuses on the first study of state-of-the-art deep video understanding models from a neuroscience perspective with more than two million regression fits.

**Voxel connectivity encoding models.** The brain is an interconnected system with local correlations within one region and global correlations across regions (Genç et al., 2016; Li et al., 2022). Few recent works explored the potential of using cortical connectivity in neural encoding models (Mell et al., 2021; Xiao et al., 2022). In (Mell et al., 2021), they proposed a model that used predefined source voxels as input to predict a target voxel and compared it to the vanilla stimulus-to-voxels prediction models as the standard neural encoding scheme. Nonetheless, these voxels-to-voxels models are not designed to take stimulus as input and define source voxels in an ad hoc manner. Our work is inspired by that direction, yet we propose a fully integrated model that learns a two-stage architecture, stimulus-to-voxels and voxels-to-voxels. More importantly, our approach takes into consideration all voxels from all visual cortex regions and learns the weighting mechanism, instead of relying on ad hoc non-learnable mechanism to define source voxels. Another work (Xiao et al., 2022) proposed an encoding approach to improve the neural predictions of high-level visual areas using the predictions of low-level visual areas. On the other hand, our method enables learning from all voxels in the same region and other regions at once within a learnable scheme to leverage inter and intra-region connectivity priors. Additionally, our scheme allows for learning from multiple regions and from the same region in one shot, while previous works mainly used connectivity to one paired region. Finally, we are the first to demonstrate the impact of encoding dynamics on utilizing these connectivity priors to improve neural encoding performance.

## 3 METHOD

In this section, we describe our environment design for the biological target and the artificial neural network target experiments. Then, we discuss the candidate video understanding models and our proposed neural encoding scheme.

### 3.1 ENVIRONMENT DESIGN

In biological neural systems encoding, we aim to study how different stimuli map to the recorded brain responses. It is usually studied within the framework of aligning and comparing deep network architectures and biological neural systems. In this case, the biological neural system is considered the target model, and the candidate deep network architecture, that extracts features from the stimuli to be mapped to the brain responses through deep regression, is considered the source model. However, it is an open question if system identification is possible and whether we can provide mechanistic understanding and insights into brain computations. Accordingly, our first goal is to answer the question: "Can we perform system identification for the underlying dynamics encoding scheme?" We use dynamics encoding to refer to the model's ability to learn from dynamic information provided in an input clip and encode it within the learned representations. To answer that, inspired by previous work (Han et al., 2023), we use the features extracted from known architectures as the target, on which we apply our regression, instead of the brain responses as an upper bound. Using a known deep neural network architecture as a target can assess how effective system identification can be and how much insight it provides when working with biological targets, i.e. the human visual cortex. Unlike previous work that focused on comparing different architectures (i.e., convolutional *vs*. transformer-based) (Zhou et al., 2022; Conwell et al., 2021b; Han et al., 2023), we go beyond that to

Table 1: List of the candidate models and their families and configurations that were used during their training. We list the backbone/s, the training datasets, and the configuration as clip length $\times$ sampling rate. For the training datasets we use ImageNet (Deng et al., 2009) (IN), Kinetics-400 (Kay et al., 2017) (K400), Charades (Sigurdsson et al., 2016) (Ch) and Something-something v2 (Goyal et al., 2017) (SSV2).

| Input | Supervision | Architecture | Network (Backbone/s - Dataset/s - Config.) |
|---|---|---|---|
| Video | Fully-supervised | Convolutional | C2D (R50-K400-8 $\times$ 8) |
| | | | CSN (R101-K400-32 $\times$ 2) |
| | | | I3D (R50-K400-8 $\times$ 8) |
| | | | R(2+1)D (R50-K400-16 $\times$ 4) |
| | | | SlowFast (R50,101-K400,Ch,SSV2-8 $\times$ 8,4 $\times$ 16) |
| | | | 3DResNet (R18,50-K400,Ch,SSV2-8 $\times$ 8,4 $\times$ 16) |
| | | | X3D (XS,S,M,L-K400-Matched Sampling Rate) |
| | | Transformers | MViT (B-K400-16 $\times$ 4,32 $\times$ 3) |
| | | | TimeSformer (B-K400,SSV2-8 $\times$ 8) |
| | | | OmniMAE finetuned (B-SSV2-8 $\times$ 8) |
| | Self-supervised | Transformers | stMAE (L-K400-8 $\times$ 8) |
| | | | OmniMAE (B,L-IN/SSV2-8 $\times$ 8) |
| Image | Fully-supervised | Convolutional | ResNet (R152,101,50,34,18-IN-8 $\times$ 8) |
| | | Transformers | ViT (B16,32,L16,32-IN-8 $\times$ 8) |
| | Self-supervised | Transformers | DINO (B-IN-8 $\times$ 8) |
| | | | MAE (B-IN-8 $\times$ 8) |

study, on a large-scale and comprehensive manner, whether models process standalone images or learn dynamics from the input video (i.e. image *vs.* video understanding models).

The second question we aim to study is: "How do families of deep video understanding models compare to biological neural systems?" Towards this, we study the identification across families of models when encoding the brain responses. Specifically, families are defined based on: (i) the input, whether models learned from single images or videos encouraging them to learn dynamics and motion, (ii) the supervision, whether they are trained in a fully supervised framework for a certain downstream task or in a self-supervised manner using unlabeled data, and (iii) the architecture, whether the model architecture relies on local convolutional operations or transformer-based global operations. While previous works focused on the architecture aspect, we argue it is even more important to look into whether the model is learning dynamics (e.g., motion) or simply using static information from a single image. Moreover, it is important to understand the impact of the supervision signal used to train the model. Throughout all the experiments we utilize Mini-Algonauts 2021 dataset (Cichy et al., 2021), in addition to providing additional results on the BOLD moments dataset (Lahner et al., 2024). In the following, we describe the design details of both the artificial neural network target environment, where the target is yet another deep network representation, and the biological target environment, where the target is the human visual cortex responses.

**Artificial neural network target environment.** We select four target models from both the image/video understanding models and convolutional/transformer-based models. Specifically, we use ResNet-50 (He et al., 2016), I3D ResNet-50 (Carreira & Zisserman, 2017), ViT-B (Dosovitskiy et al., 2021) and MViT-B (Fan et al., 2021b). ResNet-50 and I3D ResNet-50 are convolutional models, while ViT-B and MViT-B are transformer-based models. On the other hand, I3D ResNet-50 and MViT-B are video understanding models, while ResNet-50 and ViT-B are image understanding models. For each target, we use other models as source. The representations are extracted from each target model as detailed in Appendix A, with input videos from Mini-Algonauts. A dimensionality reduction is conducted on these representations for computational efficiency reasons.

**Biological target environment.** In the biological target set of experiments, our target is the brain responses. In this case, we use the public fMRI datasets from Mini-Algonauts and BOLD moments (Cichy et al., 2021; Lahner et al., 2024). In each of these datasets, we use the training set (consisting of 1000 videos) and perform cross-validation over four folds. Both datasets provide fMRI recordings of ten subjects who watched short video clips of three seconds average duration with three repetitions. Each video and voxel in the brain per participant was represented by a single activation value processed and averaged across time (Cichy et al., 2021; Lahner et al., 2024). In the Mini-Algonauts 2021 dataset, we use the brain responses from nine regions of interest of the visual cortex, these are

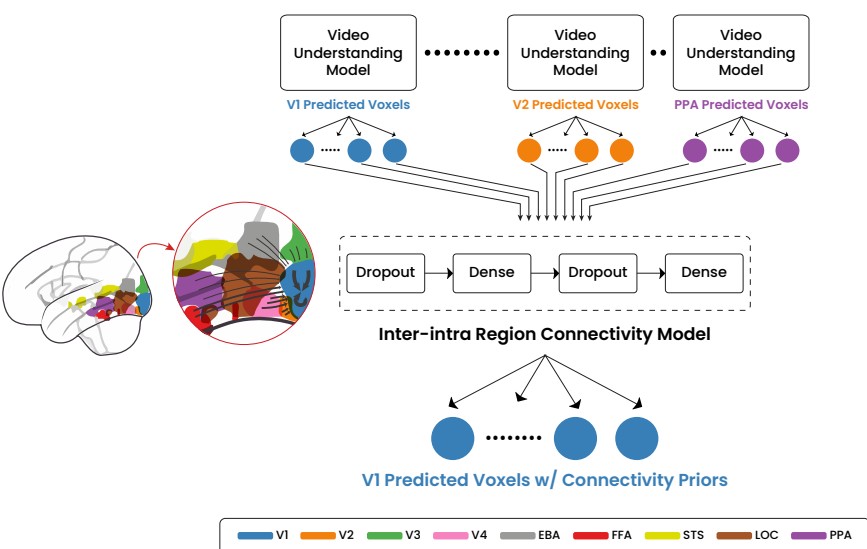

Figure 1: Architecture of our dynamics-based neural encoding with inter-intra region connectivity priors. Our fully integrated model learns the connectivity from all regions voxels (inter-region) and all voxels in the same region (intra-region). We only show one target region, V1, as an illustration where we use the same mechanism across all regions. Note the thickness of the arrows, in our visual cortex illustration, indicates the degree of connectivity corresponding to the computed in Fig. 5c.

across two levels: (i) early and mid-level visual cortex (V1, V2, V3, and V4), and (ii) high-level visual cortex (EBA, FFA, STS, LOC, and PPA). The early and mid-level visual cortex regions are concerned with lower-level features such as orientations and frequencies, while the high-level ones are concerned with semantics in terms of objects, scenes, bodies, and faces. We also conduct the same study on BOLD moments with a comprehensive 46 cortical regions that is described in the appendix B.1. The datasets we use are provided at Repetition Time (TR) one second and have shown sensitivity to the temporal ordering of information in the video stimuli (Lahner et al., 2024). This motivates our choice of the two datasets when studying video understanding models and their relation to the visual cortex when encoding dynamics.

## 3.2  CANDIDATE MODELS

Here we describe the candidate models that are used in our experiments. We choose to run our experiments on more than 35 source models, listed in Table 1, with their model families and configurations. Video understanding models include C2D (Li et al., 2019), CSN (Tran et al., 2019), I3D (Carreira & Zisserman, 2017), R(2+1)D (Tran et al., 2018), SlowFast, the Slow branch (3D ResNet-50) (Feichtenhofer et al., 2019), X3D (Feichtenhofer, 2020), MViT (Fan et al., 2021b) and TimeSformer (Bertasius et al., 2021). Self-supervised video understanding models, stMAE (Feichtenhofer et al., 2022) and OmniMAE (Girdhar et al., 2023) are used as well. We mainly focus on the models that showed state-of-the-art performance in action recognition. Single image understanding models include ResNets (He et al., 2016), ViTs (Dosovitskiy et al., 2021) and the self-supervised DINO (Caron et al., 2021) and MAE (He et al., 2022). Families of models are categorized based on the input, supervision and architecture type as discussed earlier. We detail the number of layers and which layers are sampled for each deep network used in Appendix A.

## 3.3  ENCODING TECHNIQUE

Inspired by the recent work (Zhou et al., 2022), we use a layer-weighted region of interest encoding that takes the hierarchical nature of deep networks into consideration. Initially, we sample the frames from a video to obtain the input clip. For image understanding models we extract features per frame, while for video ones we extract features from the entire clip at once. Then, we pre-process the input features from the different layers of a candidate model by averaging the features on the temporal dimension. This is followed by performing sparse random projection (Li et al., 2006) for

dimensionality reduction and computational efficiency reasons. Assume input features for layer, $l$, after dimensionality reduction as, $X_l \in \mathbb{R}^{C \times 1}$, with $C$ features. We learn the weights of one fully connected layer to provide the predictions of the voxels of one region of interest in the visual cortex as, $\hat{Y}_l = W_l X_l$. Where $W_l \in \mathbb{R}^{N \times C}$, $\hat{Y} \in \mathbb{R}^{N \times 1}$ and $N$ is the number of voxels in the region of interest. Instead of a simple ridge regression, we learn a weighted sum of the predictions of all layers and use the following loss to train our regression model,

$$\mathcal{L} = \|Y - \sum_{l=1}^{L} \omega_l \hat{Y}_l\|_2^2 + \beta_1 \sum_{l=1}^{L} \|W_l\|_2 + \beta_2 \|\omega\|_1, \quad (1)$$

where $\omega_l$ is a learnable scalar weight for layer, $l$, and, $\omega$, is the vector of weights. Each $\omega_l$, controls the contribution of layer, $l$, to the final regression of the region of interest, and $\beta_1, \beta_2$ are hyper-parameters of the regularization. We use L1 regularization for the layer weights to enforce sparsity. This encoding scheme avoids unnecessary assumptions that there is a one-to-one alignment between layers and visual brain regions of interest. Accordingly, this encoding scheme allows for more complex interactions among the layers and the brain regions of interest.

### 3.4 INTER-INTRA REGION CONNECTIVITY PRIORS

In this section, we present a novel encoding scheme on top of the best-performing video understanding models by fully integrating the neural encoding with inter- and intra-region voxel connectivity priors. An overview of our architecture is shown in Fig. 1. The input video stimuli go through the source video understanding model to extract multiple layers features as described in Sec. 3.3, followed by the connectivity module which takes the concatenated voxels of the nine visual regions as input. This module consists of four layers: two fully connected coupled with L2 regularization and two dropout ones. Finally, the output of the model is the predicted voxels of a single visual region. We train our model in a two-stage fashion, where we train the standard neural encoding scheme without connectivity following Eq. 1, followed by training the connectivity module using standard L2 regularized regression loss. First, we train regression without connectivity where the video stimulus is used as input to the source video understanding model from which representations are extracted and used for predicting voxel activations. Second, the inter/intra region connectivity module is then applied by using the voxels of the nine regions as inputs. In the training phase ground-truth voxel activations are used as input, where the target is to learn the connectivity between the voxels of the target region itself, i.e. intra-connectivity, and between voxels of the target region and the other regions, i.e., inter-connectivity. However, in the inference phase, the input to the model is the predicted voxel activations from all regions.

## 4 EXPERIMENTAL RESULTS

### 4.1 IMPLEMENTATION DETAILS

In this subsection, we describe our experiment design and implementation details. The input clips to all models are constructed based on a sampling rate that corresponds to the sampling rate used during its training for video understanding models. As for image understanding models we use the default sampling rate of eight. The clip length is computed based on the sampling rate and the input video length and changes according to the video length.

Before training the regressor, a hyperparameter tuning is conducted using two-fold cross-validation on the fold's training set of the first subject, following previous work (Zhou et al., 2022). The two main hyper-parameters we tune are the $\beta_1, \beta_2$, using grid search $\beta_1 \in \{0.1, 1, 10\}$ and $\beta_2 \in \{1, 10, 100\}$. Moreover, an early-stopping strategy is employed through the hyperparameter tuning and training phases. The main metric used throughout the experiments is the average Pearson's correlation coefficient across all voxels within a specific region in the brain. All results are averaged over the subjects. We conduct experiments on four folds and report the average and standard deviation. We report the statistical significance across families of models using Welch's t-test. The biological target setup, in which the human visual cortex is the target, is conducted on the Mini-Algonouts 2021 dataset (Cichy et al., 2019), additional results on BOLD moments dataset is provided in Appendix B.1. In each fold of the randomly selected four folds we used in our experiments, the 1,000 videos are

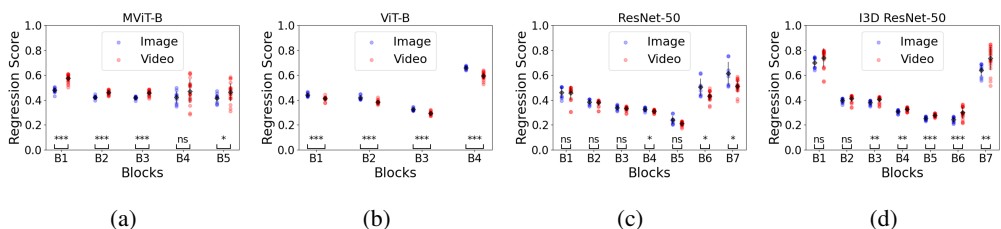

(a)            (b)            (c)            (d)

Figure 2: Artificial neural network target experiments showing regression scores as Pearson's correlation coefficient of image (blue) *vs.* video (red) model families on four target models; (a) MViT-B, (b) ViT-B, (c) ResNet-50, (d) I3D ResNet-50. We show the regression on the target network output features from their respective blocks, B0-7. Statistical significance is shown at the bottom as 'ns' not significant, '$*, **, ***$' significant with p-values $< 0.05, 0.01, 0.001$, resp. It shows higher scores for the model family corresponding to the target network, especially in MViT and ViT.

split into training and testing sets as 90% and 10%, respectively. In the artificial neural network target setup, we follow the same setup for the biological target environment. All the previous models are trained on A6000 GPU with a training span of one day average per regression model.

## 4.2 CAN WE PERFORM SYSTEM IDENTIFICATION WITH RESPECT TO THE DYNAMICS ENCODING?

We first investigate the question of whether system identification of single image *vs.* video understanding models is possible. Figure 2 shows that for MViT and ViT target models, the correct family of models is better able to represent each. Specifically, we observe higher regression scores of the video understanding models for MViT and the single image one for ViT. Note that we include video understanding models that are trained on three different datasets which are Kinetics, Charades, and Something-Something v2 to ensure the results are not dependent on a certain training dataset.

When inspecting ResNet-50 as a target, we observe the mean of image understanding models is higher but is only statistically significant in three layers. When looking at I3D target model, we notice the mean of video understanding models is higher and is statistically significant except in early layers. In summary, we have demonstrated that system identification can be attained to a certain level with most of the layers showing statistical significance. Additional artificial targets experiments are provided in App. B.2. In the following we investigate the same but for the biological neural system.

## 4.3 HOW DO DEEP VIDEO UNDERSTANDING MODELS FAMILIES COMPARE TO THE HUMAN VISUAL CORTEX?

In this section, we focus on the biological target experiments, where the target model is the biological neural system, to understand the underlying mechanisms of the visual cortex. We conduct three comparisons; single image *vs.* video understanding families of models, convolutional-based *vs.* transformer-based models, and fully-supervised *vs.* self-supervised models. Figure 3a clearly demonstrates that across most brain regions, video understanding models have better capability to model the visual cortex responses than single image architectures. We believe the reason behind this is that the brain is encoding the dynamics occurring in a video similar to the encoded dynamics in video understanding models more than what occurs in single image understanding models. Additionally, it shows PPA and FFA regions with no significant difference between image and video understanding unlike EBA and STS regions relating to previous neuroscience findings (Pitcher et al., 2011; 2019). Figure 3b shows the comparison between transformer-based and convolutional-based models. It shows that convolutional models have higher regression scores across early-mid regions in the visual cortex with relatively high statistical significance. This difference decreases as we go to higher-level regions until it becomes insignificant. Interestingly, it has been noted in recent works that transformers lack the ability to capture high-frequency components (Bai et al., 2022). On the other hand, early layers in convolutional models tend to capture high-frequency components by detecting oriented gradients. This also might relate to empirical results that demonstrated vision transformers with shallow convolutional stem (i.e., three convolutional layers) perform better than the ones that directly take patches as input (Xiao et al., 2021). As such, brain modelling in the early and mid regions of

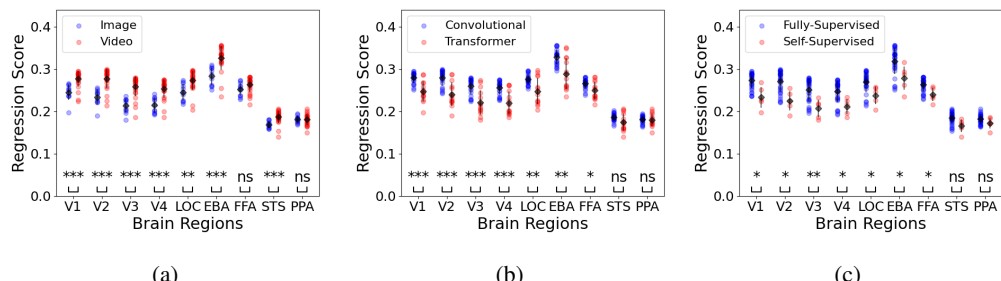

(a)                                        (b)                                        (c)

Figure 3: Biological target experiments showing regression scores as Pearson's correlation coefficient of model families on brain fMRI data. Comparison between: (a) image *vs.* video understanding models, (b) convolutional *vs.* transformer-based models and (c) fully supervised *vs.* self-supervised models. Statistical significance is shown in the bottom as 'ns' not significant, '$*, **, * * *$' significant with p-values $< 0.05, 0.01, 0.001$, resp. It shows video understanding models outperform single image ones, fully supervised outperform self-supervised ones and convolutional models surpass transformer-based ones in early-mid regions.

the visual cortex relates better to convolutional-based models, which could relate to better capturing high-frequency components. Nonetheless, we notice the best model in the transformer-based family, MViT, tends to behave similarly to convolutional ones in early-mid regions, unlike other transformer-based architectures. Additional results in Appendix B.2 are provided to confirm the previous insight. Surprisingly, Fig. 3c shows that fully-supervised models are better able to predict most of the regions than self-supervised models. Appendix B.1 provides the results of the previous study conducted on BOLD moments dataset across 46 regions, which confirms the consistency of our main findings across different datasets. Moreover, Appendix B.3 shows the model subfamilies while focusing only on the video understanding models and excluding the single image models, in addition to comparing convolutional *vs.* transformer-based models that are trained with full supervision only.

## 4.4 Fine-grained analysis

In this section, we conduct a fine-grained analysis that goes beyond families of models. We start with studying two stream *vs.* single stream architectures across three video understanding datasets. Figure 4a shows that the two-stream architectures have a better ability to model the visual cortex than single-stream ones in the low-level regions and are either better or on-par in the high-level regions. We then discuss the self-supervised learning results that showed worse regression scores in comparison to full supervision. Towards this end, we investigate OmniMAE variants (i.e., self-supervised and finetuned) and TimeSformer. Figure 4b shows that fully supervised models give better scores than the self-supervised ones across the nine regions. We leave the reasons behind this result as an open question for future research on why fully supervised models tend to align better with the cortical responses than self-supervised ones. Furthermore, we investigate which models are better at predicting the visual cortex responses. Figure 4c shows that both SlowFast, a two-stream architecture, and MViT, a multiscale vision transformer, are the best in modelling the visual cortex across the nine regions. SlowFast which is a convolutional approach is comparable to MViT with a difference that is statistically not significant across all regions. Moreover, self-supervised models (i.e., stMAE and OmniMAE) lag behind fully supervised ones. Additional analysis is provided in App. B.4 and B.5.

## 4.5 Inter-intra region connectivity

In this section, we show the results of our improved neural encoding mechanism that builds upon the best video understanding models (MViT-B and SlowFast) while incorporating intra- and inter-region connectivity. Figure 5a shows the statistically significant enhancements in prediction accuracy in both MViT-B and SlowFast models after incorporating voxel-connectivity into their predictions. These results show that integrating the intra- and inter-connectivity across the visual regions is an important component for better neural encoding. As an ablation study, we examine the performance enhancement in MViT-B in the case of intra-connectivity or inter-connectivity separately. Figure 5b shows that the full-connectivity (i.e., combining both) is either superior or on par with intra-connectivity or inter-connectivity standalone. To better understand the directional connectivity between the regions,

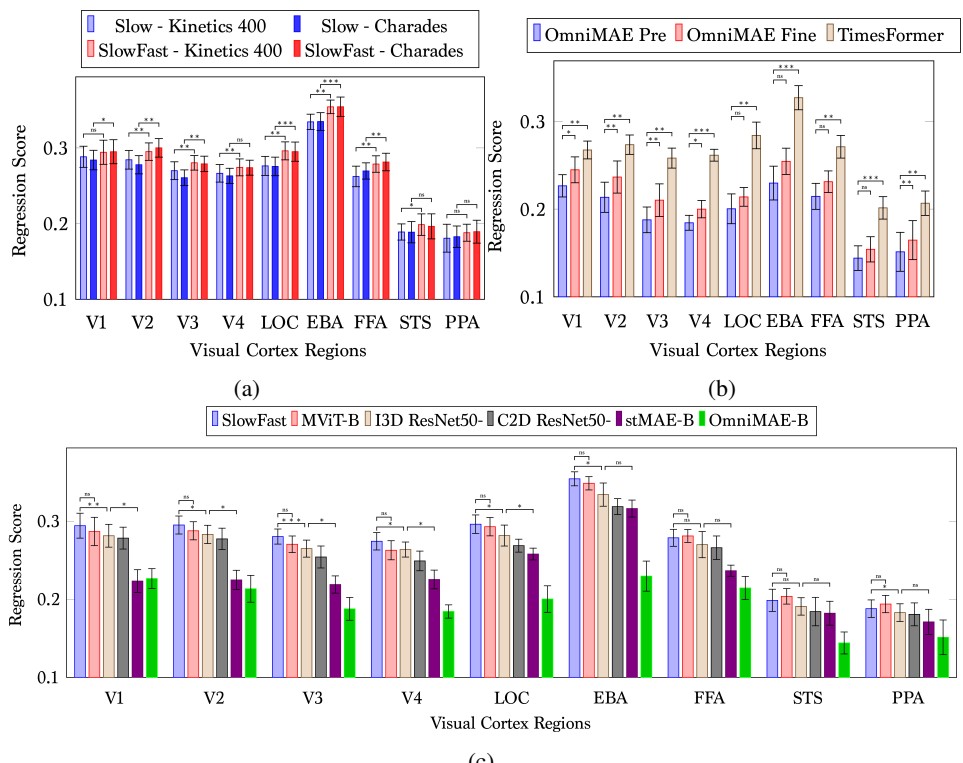

(a)

(b)

(c)

Figure 4: Fine-grained analysis of the video understanding models across the nine regions of the visual cortex showing the Pearson's correlation coefficient as the regression scores. (a) Single stream *vs.* two stream SlowFast architectures. (b) OmniMAE pre-trained in a self-supervised manner *vs.* TimeSformer and OmniMAE fine-tuned with full supervision. All models are based on ViT-B and trained on SSV2. (c) Comparison between six video understanding models. Statistical significance is shown as 'ns' not significant, '$*, **, * * *$' significant with p-values $< 0.05, 0.01, 0.001$, resp.

we analyze the average learned weights of each region as contributors to the MViT-B accuracy enhancement of each target visual region as shown in Figure 5c. It shows the following: i) the effect of one region on another is not symmetric but directional, ii) early-mid regions (V1, V2, V3, and V4) are the highest contributors to the accuracy enhancement of other early-mid regions, and the same for late-regions (LOC, EBA, FFA, STS, and PPA), iii) V4 is contributing to both early-mid and late regions, and iv) the contributions of late regions on early regions (V1, V2) are stronger than contribution of early regions on late regions which could be attributed to the top-down influence of feedback-pathways in the visual cortex (Gilbert & Li, 2013).

Finally, we perform an ablation of the improvement from intra- and inter-region connectivity prior when using an image understanding model *vs.* a video understanding one. Figure 6a illustrates the effect of inter-intra region connectivity priors on a single image understanding model (ResNet-50), where we show that it improves over the baseline with no connectivity across most of the regions with statistical significance. More importantly, we study the gain from the connectivity priors with that single image understanding model (ResNet-50) *vs.* a video understanding model that learns dynamics (MViT). Figure 6b shows that the gain from MViT is on average higher than the ResNet-50 across all the regions. When looking at the statistical significance of these results, we notice almost half of the regions show a significant gain from encoding dynamics over the single image understanding baseline. Hence, we show in these results that the dynamics encoding in video understanding models reinforces the connectivity priors. We refer to additional ablations that compare our learned connectivity to a random or an identity one in Appendix B.6, confirming the benefit of our learned connectivity.

### 4.6 SUMMARY

In summary: (i) Artificial neural network target results show that system identification between image and video understanding models is attainable to a certain level. (ii) We show that video understanding

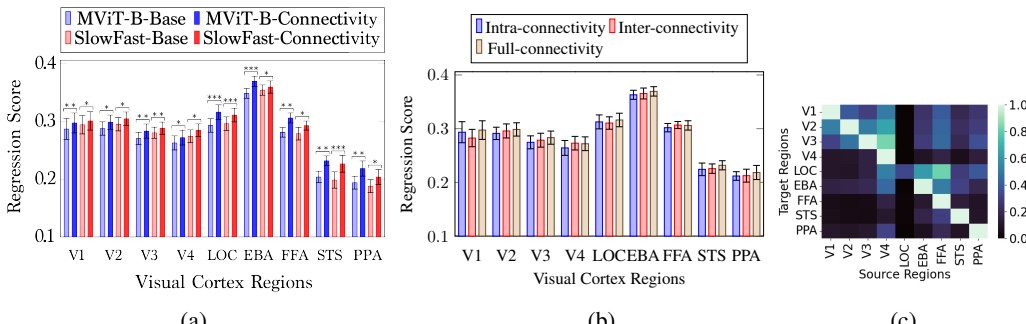

(a)                        (b)                        (c)

Figure 5: (a) Comparison of base model accuracies of MViT-B-16×4 and SlowFast and their accuracies after incorporating the intra-region and inter-region voxel connectivity showing the Pearson's correlation coefficient as the regression scores. It shows the superiority of the connectivity-based models. (b) Comparison of performance enhancement by incorporating the intra-region and inter-region voxel connectivity together or each of them separately showing the Pearson's correlation coefficient as the regression scores. It confirms the need for combining both intra- and inter-region connectivity. (c) Average weights per region contributing to the accuracy enhancement of each target visual region, showing the directional learned connectivity in our model.

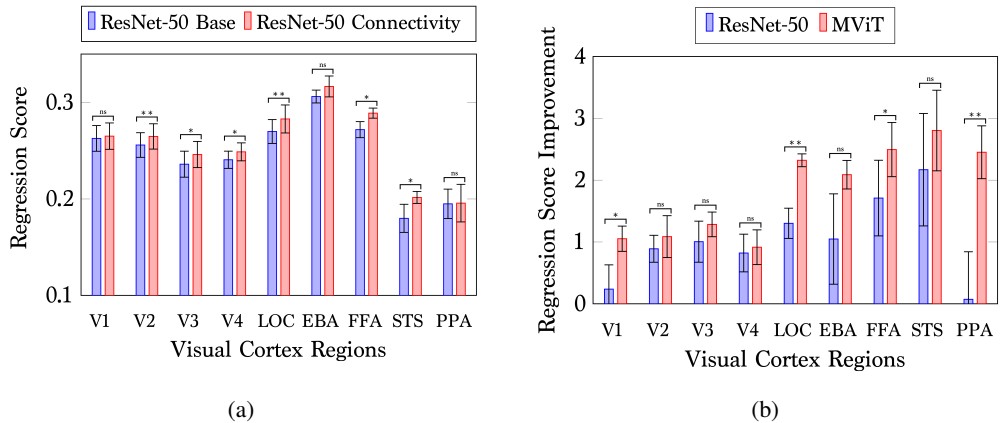

(a)                                (b)

Figure 6: (a) Comparison of base model accuracy of ResNet-50 and its accuracy with inter-intra region voxel connectivity showing the Pearson's correlation coefficient as the regression scores. (b) Comparison of improvement in the regression scores in ResNet-50 *vs.* MViT w/ connectivity priors. Difference in regression scores (i.e., Pearson correlation coefficient) is shown in a different scale after multiplying by 100 for visualization. It confirms that the dynamics-based encoding improves the benefit from connectivity priors.

models are better at regressing the visual cortex responses than image ones. (iii) We show that convolutional models predict better the early-mid regions than transformer-based ones. (iv) We show that MViT, with its multiscale processing, tends to perform similarly to convolutional models in early-mid regions. (v) We show that two-stream models perform better than the single stream. (vi) We show that models trained with full supervision surpass the ones with self-supervision. (vii) Finally, we demonstrate a better neural encoding scheme that utilizes both encoded dynamics and inter-intra region connectivity. We refer to Appendix C and D for a discussion on the limitations and impact.

## 5 CONCLUSION

This paper has provided a large-scale study of video understanding models from a neuroscience perspective. We have shown the feasibility of system identification for single image *vs.* video understanding models. Moreover, we have shown that modelling dynamics should be considered when comparing biological neural system responses to deep networks and provided insights on different families of models. Finally, we showcased the interplay of dynamics modelling and inter-intra region connectivity in neural encoding.

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

Table 2: Detailed description per architecture of the used layers.

| Architecture | # Layers | Sampled Layers |
|---|---|---|
| CSN, R(2+1)D, 3DResNet | 6 | 5 convolutional blocks and last fully connected layer |
| C2D, I3D | 7 | 5 convolutional blocks, max pooling and last fully connected layer |
| SlowFast | 12 | 5 convolutional blocks for each branch (slow & fast) and the last 2 layers combined |
| X3D | 6 | 5 convolutional blocks and last fully connected layer |
| ViT, TimeSformer | 4 | Grouped 3 blocks, each block 4 layers and last fully connected layer |
| Dino - B | 4 | 3 Grouped blocks (4 layers) and CLS token |
| ResNet | 7 | 5 convolutional blocks, average pooling and last fully connected layer |
| MViT | 5 | Grouped 4 blocks (4 layers) and last fully connected layer |
| OmniMAE, MAE | 3 | Grouped 3 blocks (4 layers) |
| stMAE | 6 | Grouped 6 blocks (4 layers) |

# A  CANDIDATE MODELS DETAILS

In this section, we provide details on the layers extracted from each of our candidate models in Table 2. Note that for transformer-based architectures instead of using all layers to be selected we rather group layers into blocks of four layers for efficiency reasons and we noticed it gave better results than learning the regression with all input layers at once. Also note that OmniMAE finetuned is trained for the action recognition task on SSV2 dataset (Goyal et al., 2017). The sampling rate used in the candidate models is the rate used to sample frames from the video clip which is computed as; number of frames in input clip = (video clip duration×frames per second)/sampling rate, as standard [1]. Video clip duration and frames per second are parameters that are based on the video stimuli, which is on average 3 seconds in our data and 30 frames per second is used. However, the sampling rate is a hyper-parameter tied to how the video understanding model was initially trained and for image understanding models we set it to eight.

# B  ADDITIONAL RESULTS

## B.1  BOLD MOMENTS DATASET (BMD) RESULTS

In this section, we evaluate on the newly released BOLD Moments Dataset (BMD) (Lahner et al., 2024), with fMRI recording of ten subjects watching 1,000 training video stimuli. The dataset provides a preprocessed version (named "Version B") with additional flexibility in the region of interest (ROIs) format, which is recommended by the dataset authors. In "Version B" of BMD, 46 regions are defined including right/left hemispheres and ventral/dorsal streams. The defined ROIs include: early visual regions (V1v, V1d, V2v, V2d, V3v, V3d, hV4), a body-selective region (EBA), an object-selective region (LOC), face-selective regions (FFA, OFA, STS), scene-selective regions (PPA, RSC, TOS), additional ROIs (V3ab, IPS0, IPS1-2-3, 7AL, BA2, PFt, and PFop), and finally the motion selective MT ROI. All these ROIs were defined in both the right and left hemispheres forming in total 46 ROIs. To further show the robustness of our results and insights, we evaluate all our models using this preprocessed version across the 46 ROIs using cross-validation on 4 folds as shown in Fig. 7. It confirms the consistency of the results previously shown in Fig. 3. Figure 7a confirms the statistically significant superiority of video understanding models compared to single-image models in the majority of cortical regions. Figure 7b confirms that convolutional-based models are better

---
[1]https://pytorchvideo.org/docs/tutorial_torchhub_inference

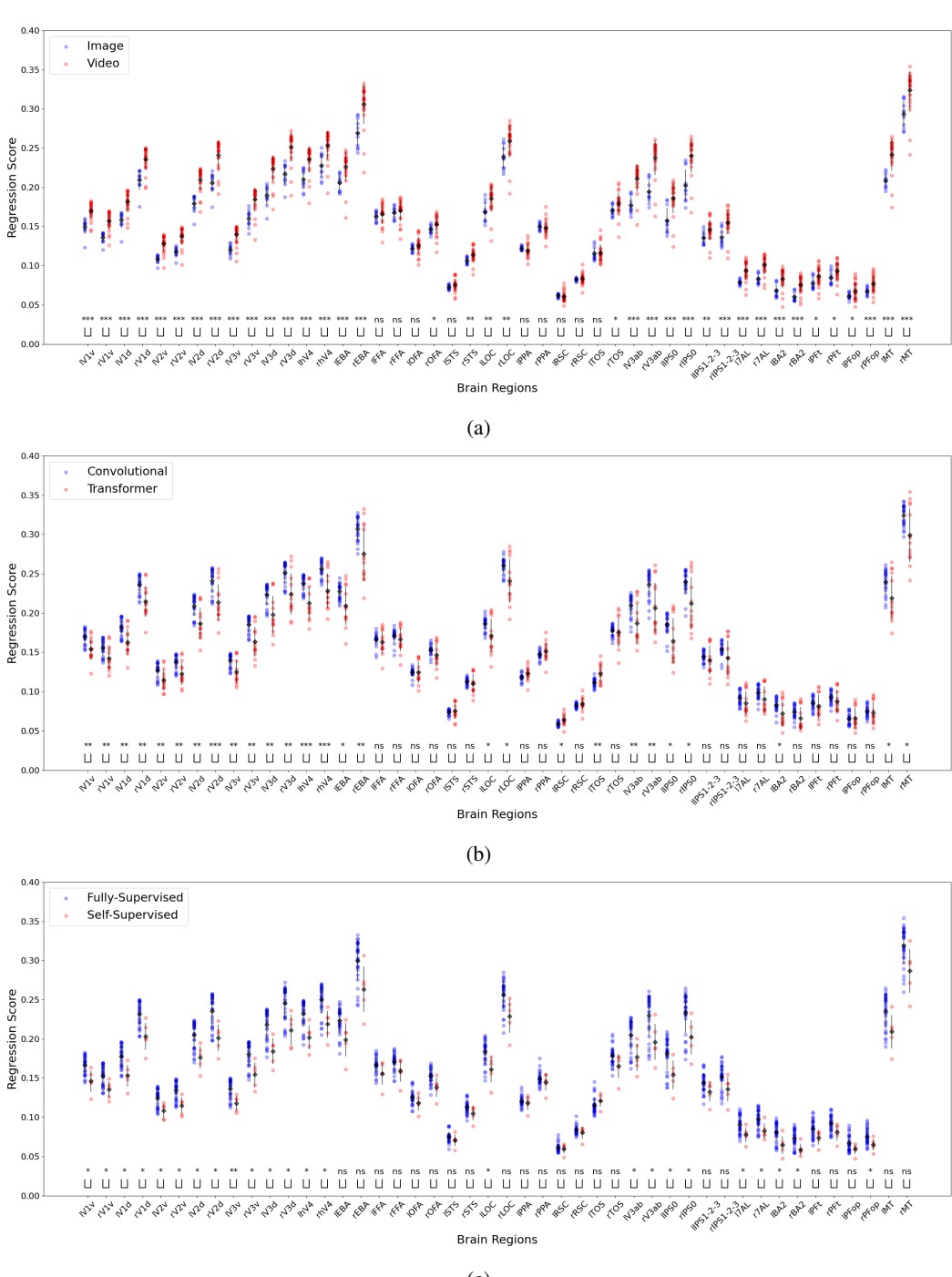

Figure 7: Biological target experiments showing regression scores as Pearson's correlation coefficient of model families on brain fMRI data using the BMD dataset. (a) Comparison of image *vs.* video understanding models, (b) comparison of convolutional *vs.* transformer-based models and (c) comparison of fully supervised *vs.* self-supervised models. Statistical significance is shown in the bottom as 'ns' not significant, '$*, **, * * *$' significant with p-values $< 0.05, 0.01, 0.001$, resp.

than transformer-based models in representing early-mid visual cortex regions. Furthermore, Fig. 7c confirms that fully-supervised models perform better than self-supervised ones in predicting the activity of most of the visual cortex regions with statistical significance.

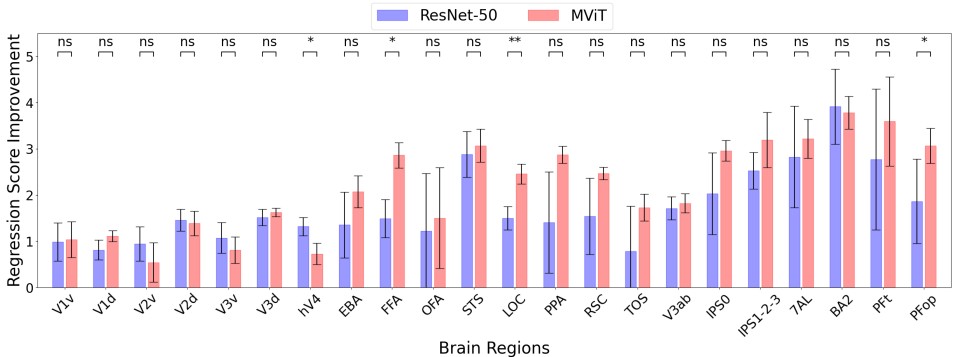

Figure 8: Neural encoding performance gain after incorporating connectivity priors in ResNet-50 *vs.* MViT-B using BMD dataset.

Finally, we ablate our proposed novel connectivity on BMD dataset. For these experiments we focus on "Version A" as it encompasses 22 regions only without splitting the hemispheres. Hence, we choose it for both computational efficiency reasons and to simplify the connectivity among the regions independent of the hemispheres. Fig. 8 shows the connectivity gains in two models, a video understanding one (MViT-B) and an image understanding one (ResNet-50). It clearly shows consistent improvement across the 22 regions for both models confirming our previous findings. More importantly, it shows on average the performance gain in MViT-B is higher than in ResNet-50 in most of the regions, which confirms the interplay between dynamics encoding and connectivity.

## B.2 ADDITIONAL ARTIFICIAL NEURAL NETWORK TARGET EXPERIMENTAL RESULTS

Although our focus in the system identification is on the ability to differentiate image *vs.* video understanding models, we provide additional results for other families of models. In Fig. 9, we show the artificial neural network target results comparing convolutional *vs.* transformer based models for four target models MViT-B 16x4, ViT-B, ResNet-50, and I3D ResNet-50, respectively. For the target model I3D, it clearly shows that convolutional models are better predictors than transformer-based ones with statistical significance across blocks except the last one. The target model ResNet50 shows that convolutional models are significantly better than transformer-based ones in the early blocks, but the differences are not significant in the late blocks. For ViT-B, the transformer-based models are significantly better than the convolutional models in all the blocks except for the first block which has a non-significant difference between the two families. This result is consistent with the results presented in (Han et al., 2023). The target model MViT-B shows a surprising result, that is confirming with previous findings in the biological target experiments as well as detailed in Section B.4, where convolutional models are better in regressing the multiscale variant than transformer-based ones with significant differences in the first two blocks. This might be related to how the multiscale ViT tends to act similarly to the convolutional models when predicting early-mid regions of the visual cortex.

In the MViT target model results showed in Fig. 2a and Fig. 9a, we did not include the MViT-32x3 variant for fair comparison between model families given that MViT-32x3 share the same architecture as MViT-16x4. However, for further investigation and understanding of MViT target model results, we re-assessed the results after adding MViT-32x3 as a source model. Figure 10 shows the results of the MViT-16x4 target model after including the MViT-32x3 source model. Figure 10a shows consistent results as shown in Figure 2a in which video understanding models are significantly better than image understanding ones. Figure 10b also shows consistent results to Figure 9a in which convolutional models are better than transformer-based models as predictors of MViT target model but with insignificant results, after adding MViT-32x3 source model to the transformers family, as anticipated.

## B.3 ADDITIONAL BIOLOGICAL TARGET EXPERIMENTAL RESULTS

We add a study of the model subfamilies in terms of (a) convolutional *vs.* transformer-based models and (b) fully *vs.* self-supervised models, but focused on video understanding models only excluding

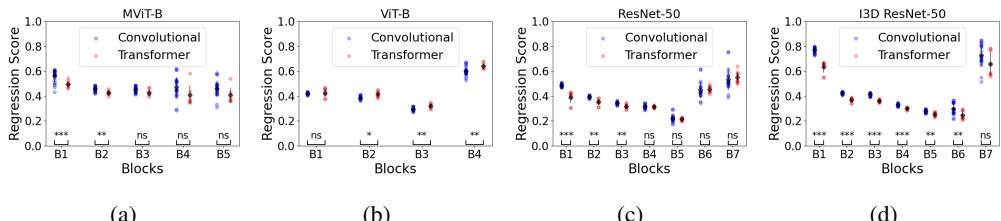

(a)      (b)      (c)      (d)

Figure 9: Artificial neural network target experiments showing regression scores as Pearson's correlation coefficient of (a-d) convolutional (blue) *vs.* transformer (red) model families on four target models MViT-B 16x4, ViT-B, ResNet-50, and I3D ResNet-50, respectively. Statistical significance is shown at the bottom as 'ns' not significant, '*, **, * * *' significant with p-values $< 0.05, 0.01, 0.001$, respectively.

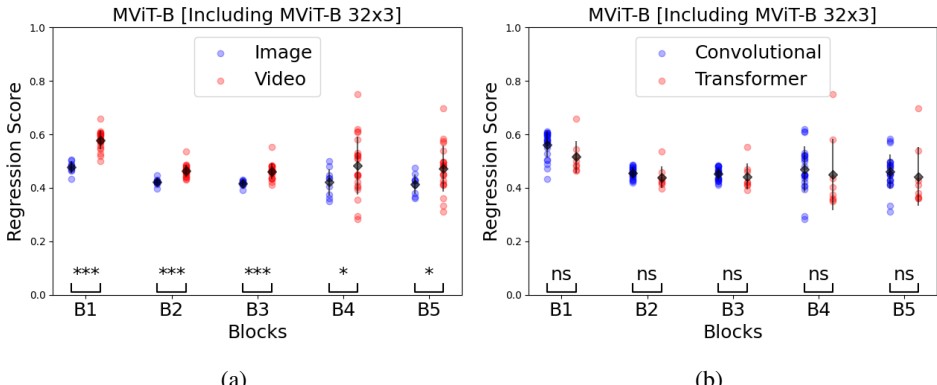

(a)             (b)

Figure 10: Artificial neural network target experiments for MViT-B 16x4 target model after including MViT-B 32x3. Statistical significance is shown at the bottom as 'ns' not significant, '*, **, * * *' significant with p-values $< 0.05, 0.01, 0.001$, respectively.

models trained with single images. As shown in Fig. 11a it shows consistently that convolutional-based models perform better in early layers as found earlier, where the first three regions show statistically significant results. Moreover, Fig. 11b shows models trained with self-supervision tend to be worse than fully supervised ones. However, note that these results use a relatively small number of self-supervised learning video understanding models. Thus, we leave it for future work to expand on this further. Finally, we study the convolutional *vs.* transformer-based models with focus on fully supervised ones excluding self-supervision. Figure 11c shows the consistency of our results where the convolutional models in the early-mid regions surpass the transformer-based ones.

## B.4 ADDITIONAL FINE-GRAINED ANALYSIS

In this section, we provide additional fine-grained analysis for both biological target and artificial neural network target environments. First, we demonstrate the layer contribution of the studied video understanding models across different regions to have a better understanding of the hierarchical nature of biological neural systems. We show the layer contribution to the regression according to the regularized encoding technique in Section 3.3. Figure 12 shows the layer contribution for three video understanding models; MViT, I3D and SlowFast (Slow and Fast branches). It clearly demonstrates that across the four early layers in these networks, there is a higher contribution to the early and mid-level regions (V1-4), and the opposite occurs as we go deeper.

Second, in Fig. 13 we compare instances of convolutional (i.e., Slow) and transformer (i.e., TimeS-former) based models. It clearly shows that especially across the early-mid regions in the brain (i.e., V1-4), the convolutional models tend to provide better regression scores. This confirms our previous insight that convolutional models are better at capturing orientation and frequencies because of their

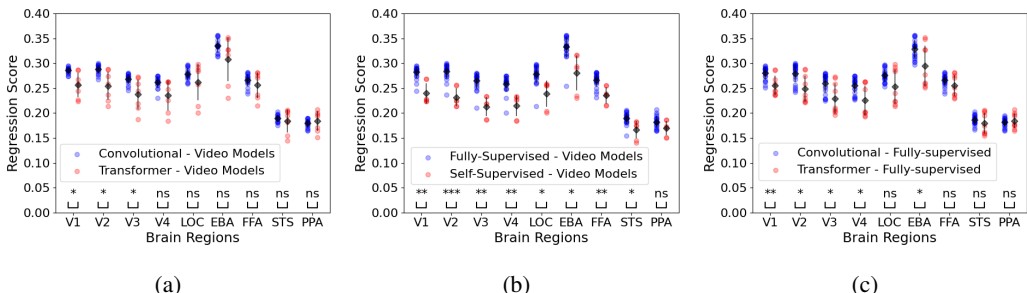

(a)  (b)  (c)

Figure 11: Biological target experiments showing regression scores as Pearson's correlation co-efficient of model families on brain fMRI data with a focus on video understanding models. (a) Comparison of convolutional *vs.* transformer-based amongst video understanding models. (b) Comparison of fully *vs.* self-supervised amongst video understanding models. (c) Convolutional *vs.* transformer-based fully-supervised models. Statistical significance is shown in the bottom as 'ns' not significant, '*, **, ***' significant with p-values $< 0.05, 0.01, 0.001$, resp.

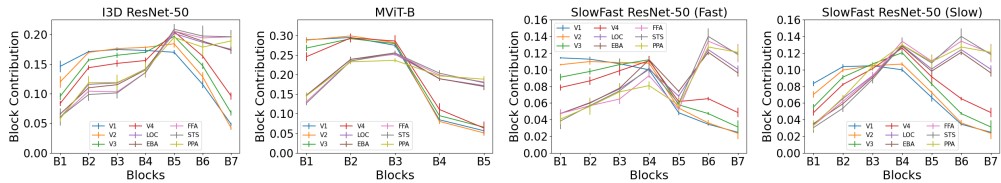

Figure 12: Layers contribution of four models for the nine regions of interest in the visual cortex.

early local connections. In later regions in the brain (i.e., LOC, EBA, FFA, STS, PPA) convolutional model (i.e., Slow) tends to act on par or less than the transformer-based model (i.e., TimeSformer).

Third, we analyze the best and worst across the models trained on single images *vs.* videos across both the artificial neural network target and biological target experiments. For the artificial neural network target experiments in Fig. 2, Table 3,4,5 and 6 show the worst (Min) and best (Max) predictors from each category (Image *vs.* Video) for target models MViT-B 16x4, ViT-B-16, ResNet-50, and I3D ResNet-50, respectively. For I3D, ResNet-50 and ViT, the results convey a simpler message that the best regressors are built with features extracted from architectures that exhibit higher similarity to the target model (i.e., convolutional/transformer) from both Video and Image understanding families. However, for MViT looking at Table 3 we see the best in the Image understanding models family is ResNet-50 which aligns with our previous insight that MViT tends to behave similarly to convolutional models, especially in the early layers.

Additionally, we show the best and worst predictors in the biological target experiments in Fig. 3 with the visual cortex regions as the target in Table 7. It shows both SlowFast and MViT are the best predictors from the video understanding family across the brain regions, with the SlowFast better at early regions similar to our previous findings as a two-stream and convolutional variant. It also shows ResNets to be the best from the image understanding family.

Fourth, we conduct an experiment comparing a randomly initialized network with respect to a trained model. The motivation behind this experiment is to confirm that our neural encoding is indeed capturing the learned representations in deep networks beyond random weights. Figure 14a shows SlowFast trained weights *vs.* random weights to confirm that the random weights provide a worse correlation coefficient than the trained ones. This is indicative that our results are not only emergent from the architecture but rather the learned representations that depend on architecture, dynamics encoding, training dataset and training signal (full/self-supervision). Finally, we conduct experiments where we show the centered kernel alignment scores following previous works (Han et al., 2023) in Fig. 14b. It shows that SlowFast, a video understanding model, surpasses ResNet-50, a single image understanding one, even when using another metric beyond the correlation coefficient of the regressed output.

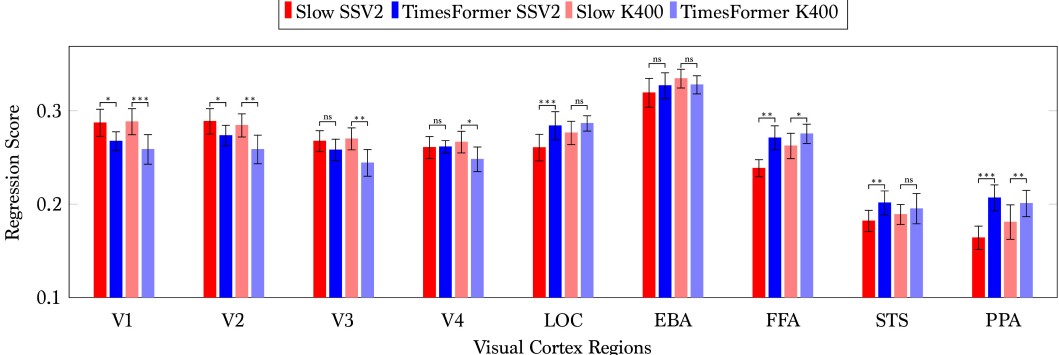

Figure 13: Fine-grained analysis comparing instances of Convolutional (i.e., Slow) *vs.* Transformer (i.e., TimeSformer) based models with different training datasets (SSv2 and K400).

Table 3: Fine-grained analysis of the artificial neural network target experiments results in Fig. 2a with target model MViT-B 16x4 showing the worst (Min) and best (Max) source model from image & video understanding models. B1-5: Blocks in the target model.

|     | Min (Video)      | Max (Video)  | Min (Image) | Max (Image)  |
|-----|------------------|--------------|-------------|--------------|
| B1  | OmniMAE-B-Fine   | MViT-B-32x3  | ResNet-34   | ResNet-50    |
| B2  | OmniMAE-B-Fine   | MViT-B-32x3  | ViT-B-16    | ResNet-152   |
| B3  | Slow-R50-SSv2    | MViT-B-32x3  | ViT-B-16    | ResNet-101   |
| B4  | Slow-R50-SSv2    | MViT-B-32x3  | ViT-B-16    | ResNet-34    |
| B5  | Slow-R50-SSv2    | MViT-B-32x3  | ViT-B-16    | ResNet-101   |

## B.5 STATISTICAL SIGNIFICANCE RESULTS

Tables 8,9a shows the pairs of models from Fig. 4a (single-stream *vs.* two-stream) and Fig. 4b (fully-supervised *vs.* self-supervised) that exhibited a statistically significant result compared to each other. Table 8 confirms that two-stream models surpass the single-stream counterpart, across eight of the nine visual cortex regions, with a statistically significant result. The table specifically shows that the superiority of the two-stream models is independent of the training dataset. In five of nine brain regions, the two-stream models were superior compared to single-stream models at two different model versions that were matched based on their training dataset (K400 and Charades). On the other hand, Table 9a confirms that fully-supervised (i.e., OmniMAE Fine and TimeSformer) surpass the self-supervised counterpart (i.e., OmniMAE Pre) across all the visual cortex regions with a statistically significant result when the three models share the same architecture base (ViT-B) and training dataset (SSv2). These results, in addition to Fig. 4b, show that TimeSformer (trained solely using full-supervision) achieved the highest regression scores followed by OmniMAE Fine (trained using both self- and full-supervision) and finally OmniMAE Pre (trained solely using self-supervision). It demonstrates that full supervision better reflects the visual cortex responses. Table 9b shows the significance results between video understanding pairs of models. As shown in Table 9b and Fig. 4c, SlowFast is statistically better than I3D in seven of the nine brain regions, I3D is statistically better than stMAE in five brain regions including four early regions of the visual cortex (V1 to V4), while SlowFast is not statistically different than MViT in any of the regions.

## B.6 ADDITIONAL RESULTS ON INTER-INTRA REGION CONNECTIVITY

In this section, we provide additional experiments and ablations on the connectivity module. First, Figure 15 shows the performance increase in neural encoding after incorporating the connectivity prior in eight additional models: SlowFast-R101, TimesFormer K400, TimesFormer SSV2, 3D ResNet-50, ViT-B-16, MViT-B-32×3, ResNet-18 and 3D ResNet-18. Second, we explore the effect of architecture as a confounding factor on connectivity performance gain. In Figure 16a, we compare the connectivity model on ResNet50 (image) *vs.* SlowFast (video) from the convolutional family. It shows on average SlowFast gain outperforms that of ResNet-50 in most of the regions. Furthermore,

Table 4: Fine-grained analysis of the artificial neural network target experiments results in Fig. 2b with target model ViT-B-16 showing the worst (Min) and best (Max) source model from image & video understanding models. B1-4: Blocks in the target model.

|    | Min (Video)       | Max (Video)       | Min (Image) | Max (Image) |
|----|-------------------|-------------------|-------------|-------------|
| B1 | MViT-B-32x3       | OmniMAE-B-Fine    | ResNet-101  | ViT-L-16    |
| B2 | SlowFast-R50-Char | OmniMAE-B-Fine    | ResNet-152  | ViT-B-32    |
| B3 | SlowFast-R50-Char | OmniMAE-B-Fine    | ResNet-34   | ViT-L-16    |
| B4 | Slow-R50-SSv2     | TimeSformer-K400  | ResNet-18   | ViT-B-32    |

Table 5: Fine-grained analysis of the artificial neural network target experiments results in Fig. 2c with target model ResNet-50 showing the worst (Min) and best (Max) source model from image & video understanding models. B1-7: Blocks in the target model.

|    | Min (Video)       | Max (Video)       | Min (Image) | Max (Image) |
|----|-------------------|-------------------|-------------|-------------|
| B1 | MViT-B-32x3       | SlowFast-R50-SSv2 | ViT-B-16    | ResNet-34   |
| B2 | MViT-B-16x4       | C2D               | ViT-B-16    | ResNet-152  |
| B3 | MViT-B-16x4       | C2D               | ViT-B-16    | ResNet-18   |
| B4 | SlowFast-R50-Char | OmniMAE-B-Fine    | ViT-B-16    | ResNet-34   |
| B5 | SlowFast-R50-SSv2 | TimeSformer-K400  | ViT-B-16    | ResNet-101  |
| B6 | Slow-R50-SSv2     | TimeSformer-K400  | ViT-B-16    | ResNet-101  |
| B7 | Slow-R50-SSv2     | TimeSformer-K400  | ViT-B-16    | ResNet-101  |

we compare ResNet50 (convolutional image) *vs.* ViT (transformer image) in Fig. 16b, it shows the transformer based model gain outperforms that of the convolutional. Similarly, in Figure 16c we compare SlowFast *vs.* MViT, with MViT showing on average higher gain. Figure 16d and Figure 16e also compares ResNet-50 (image convolutional) *vs.* 3D ResNet (video convolutional) and ViT (image transformer) *vs.* MViT (video transformer), respectively. Our results show that dynamics based connectivity performance gain is on average higher than single image ones in the majority of the regions.

Third, we compare our learned connectivity to a simple random or identity connectivity in Tables 10 and 11. In the random connectivity model, we assign random weights (using Xavier initialization) to the fully connected layers in the connectivity model. On the other hand, in the identity connectivity model, we assign identity weights (ones) to the fully connected layers. The connectivity model with identity weights represents a model where each region is learning from all the regions equally. The results clearly show that our learned connectivity surpasses these lower baselines of random or identity-initialized connectivity. Fourth, we ablate the learned connectivity heatmap from our inter and intra-region connectivity to correlation based connectivity in Fig. 17. It clearly demonstrates that our learned connectivity is quite different from the correlation based one, with interesting insights emerging on the directionaly connectivity that we discussed in the main submission.

## B.7    COMPARISON TO STATE-OF-THE-ART METHODS ON ALGONAUTS BENCHMARK

This work focuses on studying video understanding models from a neuroscience perspective through a large-scale comparison of state-of-the-art deep video understanding models to the visual cortex recordings. Additionally, we propose a novel encoding approach using inter-intra region connectivity on the top of pre-trained deep neural network models to predict visual cortex voxel activity. To study the effect of our novel encoding approach, we compared the accuracy achieved by our best model (MViT-B with connectivity priors) to the state-of-the-art (SOA) accuracies achieved per region Zhou et al. (2022). In Zhou et al. (2022), they similarly built their encoding models on pre-trained deep learning models using the layer-weighting approach and their best model varied per brain region. Accordingly, we compare our best model results with their best model per region as shown in figure 18. It shows that our novel encoding approach achieved on average higher results across all the regions except for PPA in which it is on par. This comparison shows the accuracy enhancement achieved compared to the state of the art by incorporating connectivity priors to features extracted from pre-trained deep learning models. Note that our approach has a stronger form of cross validation as we take the mean of different subjects and we compute statistics across four folds from different

Table 6: Fine-grained analysis of the artificial neural network target experiments results in Fig. 2d with target model I3D ResNet-50 showing the worst (Min) and best (Max) source model from image & video understanding models. B1-7: Blocks in the target model.

|    | Min (Video)      | Max (Video)          | Min (Image) | Max (Image) |
|----|------------------|----------------------|-------------|-------------|
| B1 | MViT-B-32x3      | CSN                  | ViT-B-16    | ResNet-34   |
| B2 | MViT-B-16x4      | SlowFast-R101        | ViT-B-16    | ResNet-18   |
| B3 | MViT-B-32x3      | C2D                  | ViT-B-16    | ResNet-18   |
| B4 | MViT-B-32x3      | C2D                  | ViT-L-32    | ResNet-18   |
| B5 | OmniMAE-B-Fine   | X3D-L                | ViT-B-16    | ResNet-50   |
| B6 | SlowFast-R50-SSv2 | C2D                 | ViT-B-16    | ResNet-101  |
| B7 | Slow-R50-SSv2    | SlowFast-R50-K400    | ViT-B-16    | ResNet-101  |

Table 7: Fine-grained analysis of biological target experiments results in Fig. 3a with target model the visual cortex regions showing the worst (Min) and best (Max) source model from image & video understanding models.

|     | Min (Video)     | Max (Video)           | Min (Image) | Max (Image) |
|-----|-----------------|-----------------------|-------------|-------------|
| V1  | stMAE           | SlowFast-R50-8x8-Char | MAE         | ResNet-18   |
| V2  | OmniMAE-B-Pre   | SlowFast-R50-8x8-Char | MAE         | ResNet-50   |
| V3  | OmniMAE-B-Pre   | SlowFast-R50-8x8-K400 | MAE         | ResNet-50   |
| V4  | OmniMAE-B-Pre   | SlowFast-R101         | ViT-L-32    | ResNet-50   |
| LOC | OmniMAE-B-Pre   | MViT-B-32x3           | ViT-B-16    | ResNet-101  |
| EBA | OmniMAE-B-Pre   | SlowFast-R101         | ViT-B-16    | ResNet-101  |
| FFA | OmniMAE-B-Pre   | SlowFast-R50-8x8-Char | ViT-B-16    | ResNet-101  |
| STS | OmniMAE-B-Pre   | MViT-B-32x3           | ViT-B-16    | ResNet-101  |
| PPA | OmniMAE-B-Pre   | TimeSformer-SSv2      | ViT-B-16    | ResNet-50   |

videos. As such, our comparison to the baseline with no connectivity that follows Zhou et al. (2022) in Fig. 5a, better reflects our improvement gains. In future work, we will further explore the effect of integrating connectivity priors with end-to-end fine-tuning of deep learning models and ensemble approaches.

### B.8 TEST SET RESULTS

In this section, we present the neural encoding performance when training on the full 1,000 training videos and evaluating on the 'Version A' BMD test set data (102 video stimuli). Figure 19 shows the encoding performance of MViT base model in comparison to MViT connectivity model after incorporating connectivity priors. It demonstrates the consistency of our conclusions when examined on the held out test set. MViT connectivity model achieved higher regression scores than MViT base model with strong statistical significance in 17 regions out of the total 22 regions. Note that statistical significance is reported across subjects in the test set.

## C  LIMITATIONS

In this section, we discuss the limitations of our work. Our current large-scale study is limited to ten subjects watching short video clips with 1,000 videos provided. We leave it for future work to explore the collection of broader datasets with longer videos up to 5-10 minutes towards the creation of foundation models for neural encoding.

## D  BROADER IMPACT

Conducting a large-scale study on neural encoding of biological neural systems in comparison to deep neural networks has multiple positive societal impacts as it can be used to drive insights into the understanding and advancement of deep networks. Our study also improved our understanding

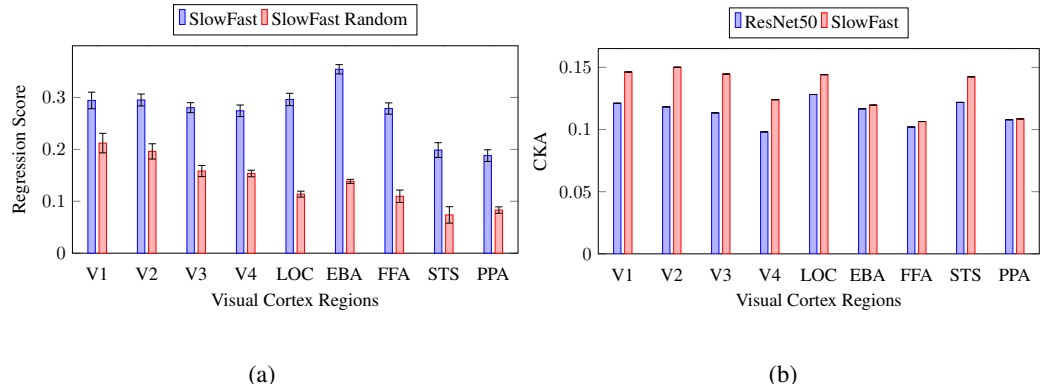

(a)                                                    (b)

Figure 14: (a) Comparison between a trained and randomized SlowFast model. (b) Centered Kernal Alignment (CKA) for ResNet50 and SlowFast models. It shows the consistency of our results using other metrics beyond the regression score.

Table 8: Statistical Significant of Slow *vs.* SlowFast (Fig. 4a, showing the pairs of models (.,.) that exhibited statistical significance. K400: Kinetics 400, Char: Charades that were used as training datasets.

|      | Stat. Sig. |
|------|------------|
| V1   | SlowFast-Char, Slow-Char |
| V2   | SlowFast-K400, Slow-K400
SlowFast-Char, Slow-Char |
| V3   | SlowFast-K400, Slow-K400
SlowFast-Char, Slow-Char |
| V4   | SlowFast-K400, Slow-K400 |
| LOC  | SlowFast-K400, Slow-K400
SlowFast-Char, Slow-Char |
| EBA  | SlowFast-K400, Slow-K400
SlowFast-Char, Slow-Char |
| FFA  | SlowFast-K400, Slow-K400
SlowFast-Char, Slow-Char |
| STS  | SlowFast-K400, Slow-K400 |

of the human visual cortex and its connectivity using our learned inter-intra-region connectivity model, which is useful in various applications. Applications for neural encoding have various benefits including providing a way for testing and understanding brain mechanisms and computations in addition to multiple neural interfaces applications. Since the neural encoding modelling and its capabilities are still being developed and relatively in its infancy, in addition to the datasets available being limited in size, we do not perceive any negative societal impact from such developed models.

## E  ASSETS AND LICENCES

We use the Mini-Algonauts Project 2021 (Cichy et al., 2021) dataset that was provided in the challenge and its development kit which is licensed under an MIT License that allows its use without restriction. The Bold Moments Dataset (BMD) including the fMRI data and stimulus metadata we used in this

Table 9: (a) Statistical Significant of Self-supervised *vs.* Fully Supervised (Fig. 4b), showing the pairs of models (.,.) that exhibited statistical significance. (b) Statistical Significant of video understanding models (Fig. 4c), showing pairs of models (.,.) that exhibited statistical significance.

(a)

| | Stat. Sig. |
|---|---|
| V1 | OmniMAE Pre, OmniMAE Fine
OmniMAE Pre, TimeSformer |
| V2 | OmniMAE Pre, OmniMAE Fine
OmniMAE Pre, TimeSformer |
| V3 | OmniMAE Pre, OmniMAE Fine
OmniMAE Pre, TimeSformer |
| V4 | OmniMAE Pre, OmniMAE Fine
OmniMAE Pre, TimeSformer |
| LOC | OmniMAE Pre, TimeSformer |
| EBA | OmniMAE Pre, TimeSformer |
| FFA | OmniMAE Pre, TimeSformer |
| STS | OmniMAE Pre, TimeSformer |
| PPA | OmniMAE Pre, OmniMAE Fine
OmniMAE Pre, TimeSformer |

(b)

| | Stat. Sig. |
|---|---|
| V1 | SlowFast-R50-8x8, I3D
stMAE, I3D |
| V2 | SlowFast-R50-8x8, I3D
stMAE, I3D |
| V3 | SlowFast-R50-8x8, I3D
stMAE, I3D |
| V4 | SlowFast-R50-8x8, I3D
stMAE, I3D |
| LOC | SlowFast-R50-8x8, I3D
stMAE, I3D |
| EBA | SlowFast-R50-8x8, I3D |
| PPA | SlowFast-R50-8x8, I3D |

Table 10: MViT learnable connectivity results compared to the base model without connectivity and other lower baselines, i.e., random and identity connectivity.

| Regions | Base Model | Learnable Connectivity | Random Connectivity | Identity Connectivity |
|---|---|---|---|---|
| V1 | 0.287 | 0.297 | -0.001 | 0.090 |
| V2 | 0.288 | 0.299 | 0.001 | 0.109 |
| V3 | 0.270 | 0.283 | 0.000 | 0.121 |
| V4 | 0.263 | 0.272 | 0.000 | 0.108 |
| LOC | 0.293 | 0.316 | 0.000 | 0.163 |
| EBA | 0.349 | 0.369 | 0.000 | 0.223 |
| FFA | 0.281 | 0.306 | 0.002 | 0.164 |
| STS | 0.204 | 0.232 | 0.001 | 0.108 |
| PPA | 0.194 | 0.218 | -0.002 | -0.009 |

study is available in the OpenNeuro database under accession code DS005165 and CC0 License (Lahner et al., 2024). Most of our image and video understanding models and their trained weights are used from Pytorch (Paszke et al., 2019) and Pytorch Video (Fan et al., 2021a), except for a few models that were retrieved from their publicly released codes and model weights.

Table 11: SlowFast learnable connectivity results compared to the base model without connectivity and other lower baselines, i.e., random and identity connectivity.

| Regions | Base Model | Learnable Connectivity | Random Connectivity | Identity Connectivity |
|---------|-----------|------------------------|---------------------|----------------------|
| V1 | 0.294 | 0.301 | 0.001 | 0.101 |
| V2 | 0.295 | 0.305 | -0.001 | 0.121 |
| V3 | 0.280 | 0.288 | -0.001 | 0.129 |
| V4 | 0.274 | 0.285 | -0.002 | 0.127 |
| LOC | 0.296 | 0.311 | -0.001 | 0.156 |
| EBA | 0.354 | 0.360 | -0.001 | 0.218 |
| FFA | 0.279 | 0.293 | -0.004 | 0.153 |
| STS | 0.199 | 0.227 | 0.002 | 0.097 |
| PPA | 0.188 | 0.204 | 0.000 | -0.019 |

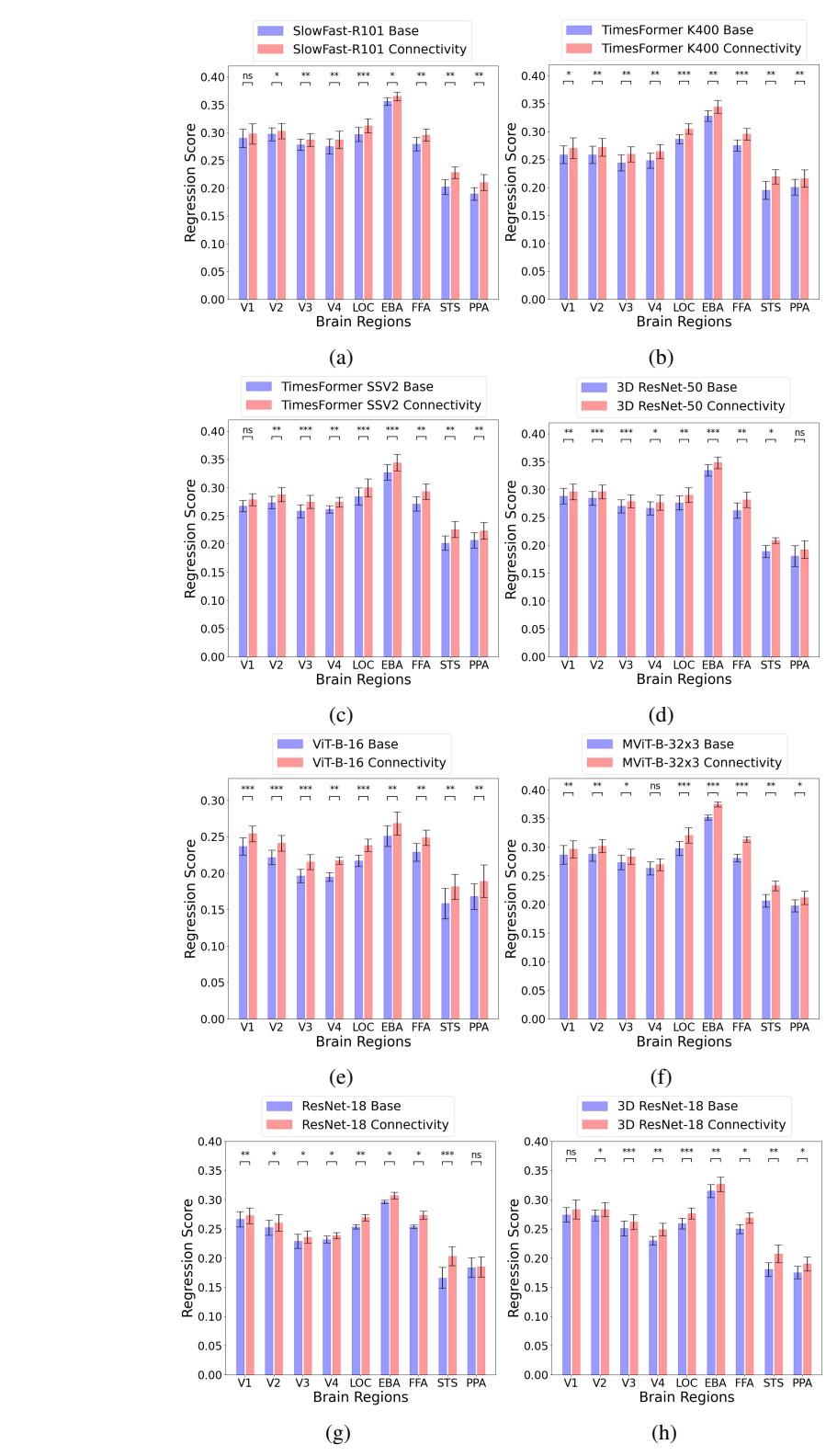

Figure 15: Comparison of base model accuracy w/o connectivity and its accuracy after incorporating the inter-intra region voxel connectivity showing the Pearson's correlation coefficient as the regression scores in (a) SlowFast-R101, (b) TimesFormer K400, (c) TimesFormer SSV2, (d) 3D ResNet-50, (e) ViT-B-16, (f) MViT-B-32 × 3, (g) ResNet-18, (h) 3D ResNet-18.

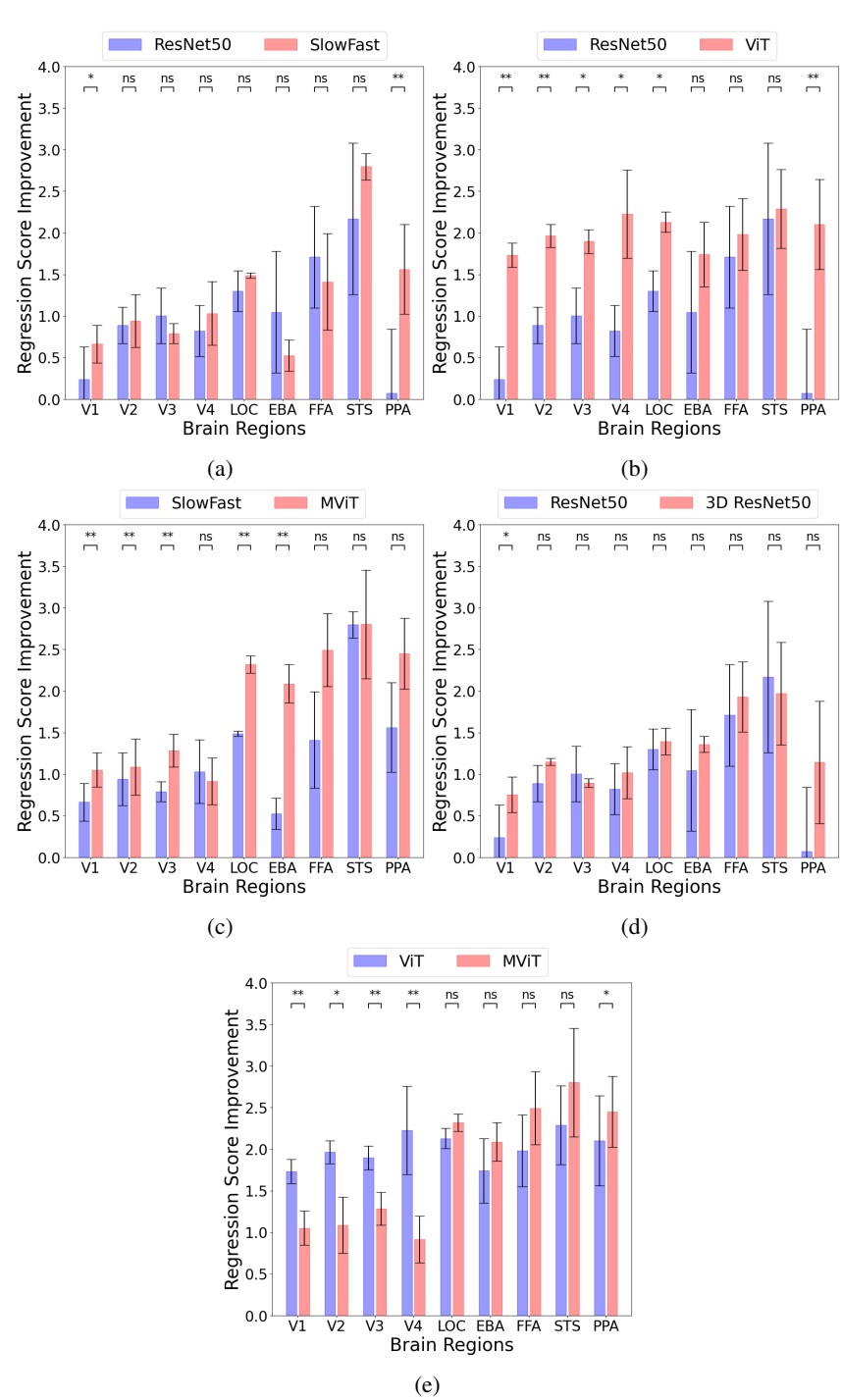

Figure 16: Comparison of improvement in the regression scores w/ connectivity priors in (a) ResNet50 *vs.* SlowFast, (b) ResNet-50 *vs.* ViT, (c) SlowFast *vs.* MViT, (d) 3D ResNet-50 *vs.* ResNet-50 and (e) ViT *vs.* MViT. The difference in regression scores (i.e., Pearson correlation coefficient) is shown in a different scale after multiplying by 100 for visualization. Statistical significance is shown as 'ns' not significant, '$*, **, ***$' significant with p-values $< 0.05, 0.01, 0.001$, respectively.

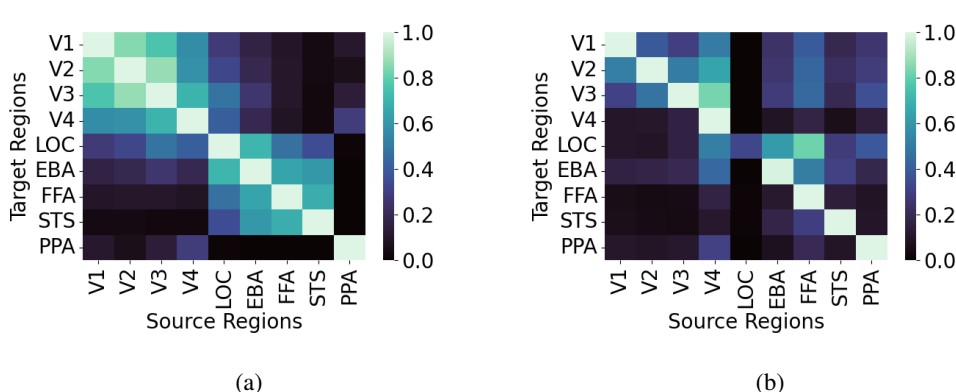

(a)                                              (b)

Figure 17: (a) Functional connectivity heatmap that is based on correlations between visual cortex regions. (b) Learned connectivity shown as average weights per region contributing to the accuracy enhancement of each target visual region. It shows the directional learned connectivity in our model.

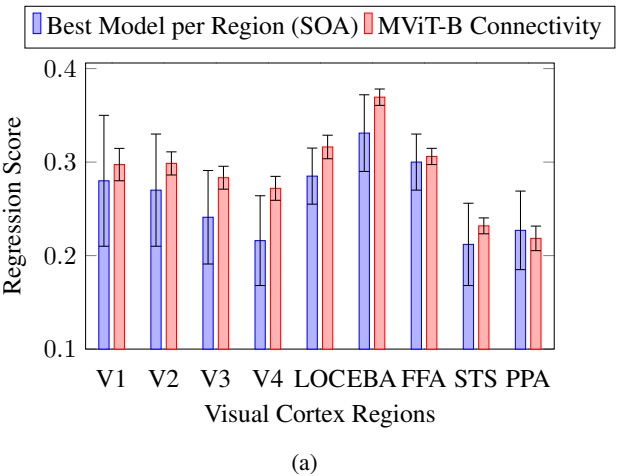

(a)

Figure 18: Comparison of MViT-B with connectivity priors to the state-of-the-art model per region as presented in Zhou et al. (2022).

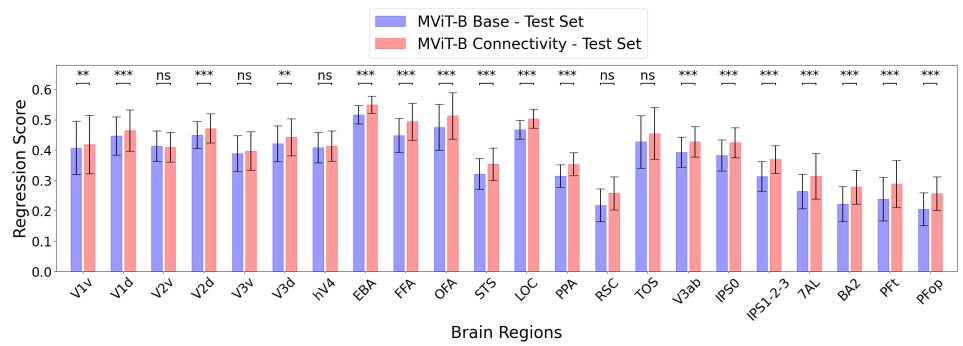

Figure 19: Neural encoding performance on test set of BMD dataset comparing MViT-B base model and connectivity model after incorporating connectivity priors. Statistical significance is shown as 'ns' not significant, '$*, **, ***$' significant with p-values $< 0.05, 0.01, 0.001$, respectively.

