# OpenReview forum: "Dynamics Based Neural Encoding with Inter-Intra Region Connectivity"
_ICLR.cc/2025/Conference — Submitted to ICLR 2025_

### Official Review · Reviewer_o1p4 · 2024-10-26

**Soundness:** 1
**Presentation:** 1
**Contribution:** 1
**Rating:** 1
**Confidence:** 5

**Summary:**

The authors use multiple neural network models to attempt to fit signals from functional magnetic resonance imaging (fMRI) obtained while participants watched short video clips from two existing datasets (Cichy et al 2021, Lahner et al 2024).

**Strengths:**

The authors have done extensive calculations with lots of computational models.
Understanding how the brain processes dynamic signals is an important question.

**Weaknesses:**

- The manuscript is poorly written and it is very hard to decipher what the authors did. There is almost no methodological information whatsoever. I list an initial list of questions that would be essential to answer when the authors submit the work to some other venue.
- It is recommended that a native speaker goes over the paper to fix grammar and clarity.
- It is also recommended that someone reads the paper with reproducibility in mind. This can help the authors understand all the information they need to provide.
- In this type of data fitting work, it is always useful to define a lower bound given by chance and an upper bound sometimes defined by computing split-half regressions across multiple repetitions of identical stimuli
- The temporal resolution of fMRI is very poor. The field has struggled with getting any meaningful temporal information about of videos. Given that the authors seem to be interested in timing, it would be useful to consider data with better temporal resolution. If the authors cannot use better data, then they should first document that there is any temporal information that can be obtained from the fMRI signals during videos.
- The authors do not say anything about eye movements. Participants typically make 2 to 4 saccades per second while watching images or video. Presumably these saccades would radically change the activity in visual cortex. It is unclear how one can make any meaningful claims about visual cortex without documenting the neural responses to each saccade.
- Before jumping into large neural networks with large numbers of free parameters, it would be useful to quantify how well the data can be explained by simple models like pixels, contrast, optic flow, etc. This is assuming that the authors first convince themselves and show that there is actual reproducible data that warrants explanation.

**Questions:**

There is almost no explanation of what the authors did here. The authors are using existing datasets and they do not need to copy and paste the entire methods section of the original papers. But they need to explicitly indicate what they have done here.
	(1) In terms of the experimental data:
	How many participants?
	How many video clips?
	What was the length of the video clips?
	How many repetitions of each video clip?
	What was the size of the image, the frame rate, and the volume of the audio?
	What was the resolution of the eye tracker used?
	What kind of video clips? What was the content? Were there cuts in the video?
	(2) In terms of the analyses,
	Saying “… we use the brain responses…” has little meaning.
	What are the units of the measurements?
	How reliable are the measurements (e.g., reproducibility across repetitions of identical stimuli)?
	Which time interval was used for the analyses? There is a mention of one second but the temporal resolution is much slower than 1 second, it would be important to document that there is indeed information at 1 second time scales.
	Which voxels were used for each region?
	Which hemispheres?
	How were the 9 regions of interest defined?
	(3) Data fitting
	What was the dimensionality of the predictors in each case?
	How was cross-validation performed?

**Details Of Ethics Concerns:**

Among the many things that are absent in the description is the notion that participants consented to the experiments, which is in the original data description.

---

> ### Author Response · Authors · 2024-11-24
> **Response to Weaknesses**
>
> **Weaknesses:**
>
> **1- The manuscript is poorly written.** We refer to all other three reviewers commending our writing style and clarity in the common response with little modifications on certain statements requested.
>
> **2- Fix grammar and clarity.** We have corrected any remaining grammar mistakes.
>
> **3- reproducibility.** While our paper already provides quite in-depth detail about our method and implementation details, we will release the code upon acceptance for full reproducibility.
>
> **4- Lower/upper bounds.** We provide in Appendix L966-971 and Fig. 14a a lower bound result which is using random weights SlowFast architecture instead of the trained weights on Kinetics/Charades/SSV2 training datasets. It clearly shows that regression using the trained model outperforms the model with random weights with a considerable margin. Additionally, the simulated experiments, i.e. with artificial neural network as target in FIg. 2, act as an upper bound for our regression since we know the ground-truth model type beforehand. Thus, it can assess the regression on biological targets and how informative it is.
>
> **5- The temporal resolution of fMRI is very poor.** We refer to Lahner et. al. (Nature Comms 2024) that confirmed the existence of degradation in results with shuffling the frames, hence the time aspect is important in the dataset we use. This is clarified in L49-53.
>
> **6- Eye movements.** We follow standard practices from a multitude of literature work that do not necessarily consider this aspect (Lahner et. al. 2024, Zhou et. al., …). We leave it for future work to consider this aspect but we do not see it as a reason for rejection.
>
> **7- Simple models.** We refer to Lahner et. al. 2024 that already conducted this in their motion energy model which is the simplest model indeed suitable for video stimuli. Our focus is rather on deep video understanding models within a large-scale comparison where we build on the aforementioned study, since the simple model has already been explored.

---

> ### Author Response · Authors · 2024-11-24
> **Response to Questions**
>
> **Questions:**
>
> **1- questions on dataset details.** Most of these were provided in our submitted manuscript, some were left out for space constraints while referring to the original dataset paper (Lahner et. al. 2024). We refer to its location in the revised manuscript in the following.
> **How many participants?** 10 subjects L213.
> **How many video clips?** 1000 in training set L212.
> **What was the length of the video clips?** short videos on average 3 seconds L213.
> **How many repetitions of each video clip?** 3 repetitions L213.
> **What was the size of the image, the frame rate, and the volume of the audio?** Image resolution is 268×268 pixels, frame rates range from 15 to 30 frames per second (mean = 28.3), and no audio. These fine-grained details are left to the original dataset paper, Lahnet et. al. 2024, for space constraints.
> **What was the resolution of the eye tracker used?** We mainly use fMRI recording of the human brain provided from Algonauts and BMD datasets, we do not use eye tracker data.
> **What kind of video clips?** Natural videos of various scenes and actions sampled from the Memento10k dataset Newman et al. (2020), as documented in Lahner et. al. 2024.
>
> **2- In terms of the analyses, Saying “… we use the brain responses…” has little meaning. What are the units of the measurements?** We mean by brain responses the fMRI recorded voxel activation values.
> **How reliable are the measurements (e.g., reproducibility across repetitions of identical stimuli)?** As mentioned in Lahner et. al. 2024, they used “both univariate and multivariate metrics to evaluate the reliability of BMD relevant. The univariate framework focuses on local information at a single voxel scale, and the multivariate analysis framework emphasizes the distributed nature of information in population codes.”
> **Which time interval was used for the analyses?** We follow the Mini-Algonauts dataset that averaged the time interval from 5 to 9 seconds in the fMRI recorded.
> **Temporal resolution.** We refer to L245 using one second which is interpolated from the original data as Lahner et. al. 2024 explained in their original work.
> **Which voxels were used for each region? How were the 9 regions of interest defined?** We follow the definitions of the regions the original dataset provides in terms of voxels definition for Algonauts and BMD. For Algonauts these are the Ventral stream regions of interest that were extended later in BMD to include the dorsal as well and additional regions.
> **Which hemispheres?** In Algonauts we do not differentiate hemispheres following their dataset, while in BMD we report in both left and right, Appendix B1, for Ver. B while Ver. A provides combined data across both hemispheres.
>
> **3-  Data fitting What was the dimensionality of the predictors in each case?** It differs from one neural network to another that is used in the feature extraction. Nonetheless, we apply sparse random projection on these features. The output from this dimensionality is on average 5000 features that is used as input to the regression model, which is then used to predict the voxels activations per region. The number of voxels per region differs according to the region itself.
> **How was cross-validation performed?** Cross validation across four folds where each fold splits the dataset into 90% training and 10% test is conducted, L315-316. We then report average and variance.

---

> > ### Comment · Area_Chair_mDze · 2024-11-25
> >
> > Dear Reviewer,
> >
> > The authors have provided their responses. Could you please review them and share your feedback?
> >
> > Thank you!

---

> > > ### Comment · Reviewer_o1p4 · 2024-11-26
> > >
> > > -- If the temporal resolution of fMRI signals is ~5 seconds (e.g., Logothetis 2001) and you indicate that the videos were 3 seconds long (approximately), what can be said about dynamic videos with such data? Which frame or part of the video do the signals map onto? It would be great to show single-trial fMRI responses to assess whether the signals have any relationship to the video content. Unfortunately, with 3 repetitions,  and given the very low signal fidelity of fMRI, it will be difficult to make any rigorous statement, but at least, it would be essential to show single trials and consistent responses in the fMRI signals to any video content.
> > > It seems that this paper inherits many problems from previous work. The answer to many questions is: "see Lahner et al 2024". This seems like a way of forward-propagating errors in the field.
> > >
> > > -- Stimulus. "Image resolution is 268×268 pixels...."
> > > Pixels are not a useful metric. Depending on the resolution of the screen and size of the screen, this amounts to completely different viewing sizes. What kind of screen was used, what was the resolution, what was the distance to the screen, what was the size of the video? How did you deal with the different frame rates?
> > >
> > > -- Eye tracking. "We mainly use fMRI recording ... "
> > > How did you control for eye movements if you did not measure eye movements? Were participants fixating or were they allowed to freely view the images? How different were the eye movements among the 3 repetitions? People typically make about 3 saccades per second. Given the poor temporal resolution of fMRI (~5 seconds), each frame lasting about 30-60ms (not clear from the description), eye movements every ~333 ms (presumably), it is essential to show that the fMRI data can capture any meaningful signal. This would require documenting single trial responses (ideally locked to each saccade), and showing that there are reliable responses.
> > >
> > > ICLR suggests including a section on reproducibility. The first step would be to document how reproducible the signals are for those 3 repetiitions. In future experiments, it would be better to have more repetitions too. But at least within those 3 repetitions, it would be essential to show that the data are reproducible.
> > >
> > > -- "We mean by brain responses the fMRI recorded voxel activation values."
> > > I know that this is what you meant. I suggest reading Logothetis 2001 as a great primer to understand the complexities of interpreting fMRI data. I also strongly suggest showing real data with real units (e.g. distances are measured in units of meters or multiples or submultiples thereof).
> > >
> > > -- How reliable are the measurements (e.g., reproducibility across repetitions of identical stimuli)?
> > > "As mentioned in Lahner et. al. 2024,...”
> > > Where do you show single trials and reproducibility of the measurements, especially with the concerns about eye movements and the dynamics alluded to above?
> > >
> > > -- Time intervals. "We follow the Mini-Algonauts... "
> > > 5 seconds, 9 seconds, something in between? What did you do? How did you account for the video dynamics and eye movement dynamics?
> > >
> > > -- Data fitting What was the dimensionality of the predictors in each case?
> > > "It differs from one neural network to another..."
> > > Reproducibility is critical. Step 1 is to report what you did, including the number of dimensions used for fitting and the number of voxels in each case. Given the problems with the data alluded to above, this is essential to assess the extent to which these data fitting efforts make sense.
> > >
> > > -- How was cross-validation performed? "Cross validation across four folds ...."
> > > The issue is how to evaluate the extent of overfitting giving the large number of features used for data fitting (see question above). How similar are the training data and the test data? There are many ways to assess similarity, starting with pixel level differences (typically not a good metric for "higher visual order" questions and even less so for higher cognitive questions), to visual features, to dynamic visual features, to aspects of the video content and narrative in the case of highly uncontrolled and complex datasets such as the videos used here.
> > > To the extent that there is some sort of reproducible signal across the 3 repetitions despite the lack of control for eye movements and for the complex video dynamics, it is not clear what this signal could relate to (low level contrast or emotions triggered by the video content or anything in between?). Assuming that there is some such unclear putative signal that relates to the videos (not shown in this paper or in the original Lahner et al 2024 work that there is any such reproducible signal), then the question would be whether how the cross-validation procedure separates along such dimensions or not. An initial step in this direction would be to document the differences between training and test sets along all the dimensions that one could possibly imagine relate to such putative signals.

---

> > > > ### Author Response · Authors · 2024-11-28
> > > > **Round 2 Response - Part I**
> > > >
> > > > ***We want to reiterate to the reviewer that the majority of the inquiries are on the dataset we are using which is already a well established peer-reviewed Nature Communications 2024 work. We are not forward propagating anything beyond the fact that fine-grained details on datasets are referred to the original work as standard.*** Our contribution is not the dataset but rather the large-scale comparison and novel neural encoding, so any inquiries in these two would be relevant to us. Nonetheless, we try to address the reviewer’s concerns as possible.
> > > >
> > > > **About dynamics encoding** The only evidence we require to confirm whether BMD and consequently Algonauts is sufficient for our dynamics encoding study is their sensitivity to frame shuffling which was shown to degrade neural encoding performance in (Lahner et. al. 2024). We here refer to the relation between frame shuffling and dynamics encoding as quite standard from a multitude of video understanding literature (Kowal et. al. 2022, Li et. al. 2021, to cite few). All other aspects, like resolution, eye tracking data and so forth might have an impact, but are out of our study focus.
> > > >
> > > > *Kowal, Matthew, et al. "A deeper dive into what deep spatiotemporal networks encode: Quantifying static vs. dynamic information." CVPR. 2022.*
> > > >
> > > > *Li, Jun, and Sinisa Todorovic. "Action shuffle alternating learning for unsupervised action segmentation." CVPR. 2021.*
> > > >
> > > > **“what can be said about dynamic videos with such data?”** For both Algonauts and its extension BMD, the data is recorded using the 3T Trio Siemens scanner at ***TR 1.75s not 5s***. Then they use a resampling technique to ensure TR is at 1s.
> > > >
> > > > **“Which frame or part of the video do the signals map onto?”** They use 3 repetitions with 1s TR and each trial is for 4 seconds. Identifying the exact frame depends on the framerate and so forth but as mentioned earlier dynamics was already proven to be captured.
> > > >
> > > > **“ It would be great to show single-trial fMRI responses to assess whether the signals have any relationship to the video content.”** Again we refer to the shuffling experiments to confirm the existence of a relationship between video content and fMRI responses. Additionally, we report the comparison between MViT with and without connectivity on the test set of BMD with 10 repetitions confirming our conclusions and going beyond 3 repetitions in the training data here and Fig. 19 in revised manuscript.
> > > >
> > > > results are sorted according to the regions as: V1v, V1d, V2v, V2d, V3v, V3d, hV4, EBA, FFA, OFA, STS, LOC, PPA, RSC, TOS, V3ab, IPS0, IPS1-2-3, 7AL, BA2, PFt, PFop. Significance of connectivity surpassing without connectivity highlighted using ^.
> > > >
> > > > ***MViT w/o connectivity: [0.407, 0.447, 0.414, 0.450, 0.389, 0.421, 0.409, 0.516, 0.449, 0.475, 0.321, 0.468, 0.314, 0.218, 0.427, 0.392, 0.383, 0.313, 0.264, 0.222, 0.239, 0.205]***
> > > >
> > > > ***MViT w/ connectivity: [0.418^, 0.465^, 0.410, 0.471^, 0.397, 0.442^, 0.414, 0.550^, 0.494^, 0.513^, 0.354^, 0.503^, 0.353^, 0.258, 0.455, 0.427^, 0.424^, 0.370^, 0.314^, 0.278^, 0.288^, 0.257^]***
> > > >
> > > > **What kind of screen was used, what was the resolution, what was the distance to the screen, what was the size of the video? How did you deal with the different frame rates?** Quoting BMD “Inside the scanner, videos were presented at the center of the screen subtending 5 degrees of visual angle and overlaid with a central red fixation cross (0.52 degrees of visual angle). Subjects were instructed to focus on the fixation cross for the duration of the main experiment” There are certain details we can not provide and as mentioned earlier has little impact on our study.
> > > >
> > > > **Eye tracking. “Were participants fixating or were they allowed to freely view the images?”**
> > > > Overlaid with a central red fixation cross (0.52 degrees of visual angle). Subjects were instructed to focus on the fixation cross for the duration of the main experiment. Participants reported luminance changes of the fixation cross occurring irregularly between videos with a button press and did so reliably (hit rate 0.964 ± 0.014 (mean ± SD))
> > > >
> > > > **“ICLR suggests including a section on reproducibility.”** We believe the reviewer is conflating ICLR definition of reproducibility for our core work with that of the dataset. We have documented all our implementation details and will release the code upon acceptance to ensure reproducibility. Beyond this is out of the scope of our study which does not have the dataset as contribution.

---

> > > > ### Author Response · Authors · 2024-11-28
> > > > **Round 2 Response - Part II**
> > > >
> > > > **“I also strongly suggest showing real data with real units.”** We are working on conducting neural encoding where the main evaluation metric is the regression score (Pearson’s correlation coefficient) which is unitless. Beyond that is irrelevant to our study.
> > > >
> > > > **“How reliable are the measurements (e.g., reproducibility across repetitions of identical stimuli)? "** We refer to Lahner et. al. 2024 reliability experiments which is out of the scope of evaluating our work.
> > > >
> > > > **Time intervals.** We used preprocessed data from Algonauts and BMD as is which they report have 4s trial length, 3 repetitions with 1.75s TR and resampled to 1s. Thus, we are not doing anything in this pre-processing we are rather using the data provided as is. They modeled the response to visual events from 1-9 s from the stimulus onset to account for the hemodynamic lag, focusing from second 5 to 9, binned at 1s.
> > > >
> > > > **Data fitting What was the dimensionality of the predictors in each case?** We do not need to report the dim. Of each predictor in each model for reproducibility. Better we can refer to any preprocessing conducted and the models we were using and this should be sufficient for reproducability. We use sparse random projection following Zhou et. a. 2022 work in any of the hyper-parameters related to this dimensionality reduction technique. We have also detailed that we conduct temporal averaging.
> > > >
> > > > **How was cross-validation performed?** We reiterate again we use 90-10% splits for four folds and these splits are randomly selected per fold. Then we report statistics across the four folds (mean, std).
> > > >
> > > > **The issue is how to evaluate the extent of overfitting.** Simply reporting across multiple folds should be sufficient. Nonetheless, we report our performance on a clearly held out test set of BMD to confirm that our dynamics based encoding with connectivity prior outperforms our baseline without connectivity. Statistical analysis in the test set is conducted across subjects, since there are no folds in this case.
> > > >
> > > > **it is not clear what this signal could relate to (low level contrast or emotions triggered by the video content or anything in between?)** I think the reviewer is overlooking the shuffling experiment which is specifically tied to the ordering of the information presented, which is tightly coupled to dynamics.

---

> > > > > ### Comment · Area_Chair_mDze · 2024-11-30
> > > > >
> > > > > Dear Reviewer,
> > > > >
> > > > > The authors have provided their responses. Could you please review them and share your feedback?
> > > > >
> > > > > Thank you!

---

> > > > > ### Comment · Reviewer_o1p4 · 2024-11-30
> > > > > **Lack of responses**
> > > > >
> > > > > The authors have failed to respond to any of the previous questions. I repeat the questions below.
> > > > >
> > > > > “I also strongly suggest showing real data with real units.”
> > > > > -- We are working on conducting neural encoding where the main evaluation metric is the regression score (Pearson’s correlation coefficient) which is unitless. Beyond that is irrelevant to our study.
> > > > > What are the units of the actual measurements on which you performed your data fitting experiments? E.g. you can measure distance in meters.
> > > > >
> > > > > “How reliable are the measurements (e.g., reproducibility across repetitions of identical stimuli)? "
> > > > > -- We refer to Lahner et. al. 2024 reliability experiments which is out of the scope of evaluating our work.
> > > > > The authors keep referring to this earlier study without answering the question. I could not find any measure of data reliability in the previous study either. How reliable are the measurements (e.g., reproducibility across repetitions of identical stimuli)
> > > > >
> > > > > Time intervals.
> > > > > -- We used preprocessed data from Algonauts and BMD as is which they report have 4s trial length, 3 repetitions with 1.75s TR and resampled to 1s. Thus, we are not doing anything in this pre-processing we are rather using the data provided as is. They modeled the response to visual events from 1-9 s from the stimulus onset to account for the hemodynamic lag, focusing from second 5 to 9, binned at 1s.
> > > > > The authors failed to answer the questions. How did you control for eye movements? Given that the temporal resolution of fMRI signals is on the order of 5s or so, how can you analyze movies with frames presented at 30Hz or so?
> > > > >
> > > > > Data fitting
> > > > > -- What was the dimensionality of the predictors in each case? We do not need to report the dim. Of each predictor in each model for reproducibility. Better we can refer to any preprocessing conducted and the models we were using and this should be sufficient for reproducability. We use sparse random projection following Zhou et. a. 2022 work in any of the hyper-parameters related to this dimensionality reduction technique. We have also detailed that we conduct temporal averaging.
> > > > > I cannot understand why the authors do not want to report what they did and what the dimensionality is.
> > > > >
> > > > > How was cross-validation performed?
> > > > > -- We reiterate again we use 90-10% splits for four folds and these splits are randomly selected per fold. Then we report statistics across the four folds (mean, std).
> > > > > You did not answer the questions. How similar are the training and test data? See the previous questions for more details.
> > > > >
> > > > > The issue is how to evaluate the extent of overfitting.
> > > > > -- Simply reporting across multiple folds should be sufficient. Nonetheless, we report our performance on a clearly held out test set of BMD to confirm that our dynamics based encoding with connectivity prior outperforms our baseline without connectivity. Statistical analysis in the test set is conducted across subjects, since there are no folds in this case.
> > > > > No, this is not sufficient. Read the previous questions and answer them please.
> > > > >
> > > > > it is not clear what this signal could relate to (low level contrast or emotions triggered by the video content or anything in between?)
> > > > > -- I think the reviewer is overlooking the shuffling experiment which is specifically tied to the ordering of the information presented, which is tightly coupled to dynamics.
> > > > > This response has nothing to do with the question. It is not clear what this signal could relate to (low level contrast or emotions triggered by the video content or anything in between?)

---

> > > > > > ### Author Response · Authors · 2024-11-30
> > > > > >
> > > > > > Since none of the above questions is a reason for strong reject, nor these details have any relevance to the core of our work, we believe our previous response is sufficient.

---

### Official Review · Reviewer_MMrh · 2024-11-03

**Soundness:** 2
**Presentation:** 2
**Contribution:** 2
**Rating:** 5
**Confidence:** 5

**Summary:**

This work is developed in two parts, aiming to (1) compare the neural predictivity of video deep neural networks on video fMRI data, and (2) propose a method to improve prediction scores via utilizing brain region connectivity.
In the first part, it tests the brain predictivity of models with different architectural design and learning paradigm and contrasts to image models (in total, 26 video action recognition models and 11 image object recognition models are tested). The results indicate that video models are better predictors than image models, in early regions convolutional models are better than transformers, and supervised models are better than self-supervised. Additionally a system identification check is performed which shows that video models can predict other video models better than image models, and vice versa, based on four target models. In the second part, this work proposes training a module to consider all regions in the prediction of any single region and shows improvements on the prediction scores of two selected video models and one image model; the image model has a less pronounced improvement which the authors claim is due to the lack of dynamics.

**Strengths:**

The paper is original in the aspect of application (model-to-brain encoding) to a new domain (human video fMRI aside from image fMRI), and the writing style has good quality. Previous work is also sufficiently covered. The questions asked are important and in general this work is a step in the right direction.

**Weaknesses:**

However, there are several soundness and presentation related issues in its execution that negatively impact this reader’s opinion of the paper.

First of all, the paper reads as two rather disjoint parts that fail to come together as one clear proposition (see summary for the two parts). As such, it looks like this could be two unrelated papers, that each would need further work to be able to stand on its own. This is the case in many points throughout the paper, for example in Figure 1, the reader is lead to think that this is the main figure / main method proposed, but the extent of the results with this method are underwhelming, as are the technical details provided.

Second, there are several issues with the soundness of the methods.
* The reader is not convinced that the improvements observed in brain predictivity are caused by the learned connectivity, and not just by obtaining a more robust or less noisy embedding by considering other regions. There are also inconsistencies in the results regarding this aspect; in figure 5c it is observed that LOC does not connect with any other areas apart from itself - then how is it explained that in figure 5a performance is significantly improved with the connectivity module? Could it be that connectivity is not the reason at all for the improvement? For the learned connectivity itself, extensive comparison with findings in neuroscience literature on region connectivity is lacking.
* For the analysis in section 4.5 as a whole, it is very unconvincing that only two models are compared, instead of the 37 models initially examined in the first part. When comparing to the image models, only one image model is tested and from that the conclusion “shows the necessity for encoding dynamics to fully benefit from such connectivity priors.” is inferred, which is too strong.  Additionally in figure 6b, it is inconsistent that not even Slowfast is shown but only MViT. In the same figure, scores “in a different scale after multiplying by 100” are shown instead of percentage of improvement, not allowing to observe the improvement regardless of the models’ performance scale itself.
* The motivation to use the Algonauts challenge’s training set for the main results without evaluating on the official test set, is unclear. An even better, more obvious choice for the main results would be to use the full Bold Moments Dataset which is more refined, higher quality, and also includes the test set.
* The supervised vs. self-supervised comparison is not clean, as all the self-supervised models are transformers - thus the relationship between convolutional models and transformers is also partly transferred in that result. The comparison is also between very different numbers of models, with only 5 SSL and more than 30 SL. Additionally, throughout the paper results are shown with models mixed from different training datasets, which makes the effects of each examined comparison (e.g. architecture) and the training dataset, entangled. See Conwell et al. (2022, 2024) for an example of clean comparisons.
* The reader finds that the fine-grained analyses of figure 4 are too fine-grained to be scientifically valid. What does evaluating a single design choice for two-stream processing tells us about either the networks or the brain? Why is the specific TimesFormer architecture chosen to compare against two other (specific) architectures as representatives of supervised, self-supervised, and fine-tuned self-supervised? Furthermore, quite a lot of emphasis is put on the multiscale component of MViT, however, without an explicit ablation it is not possible to know which factor of this model contributes to its good brain predictivity. Additionally, the authors reach the conclusion that early regions relate better to convolutional models “since they better capture high-frequency components” but this is not evidence drawn from their results and rather from other publications. It is not shown that this is the reason behind their results. Further, the model MViT does not conform to this so it seems that this conclusion is too strong.

Third, there are some major problems in the paper’s presentation.
* Figure 1 has several issues. (1) It does not show any details of the Inter-intra Region Connectivity Model (layers, loss function), which is the main component proposed. (2) It shows the pipeline only at the inference mode and does not mention neither which mode is shown or their (substantial) differences. (3) In the first stage of voxel prediction (encoding) it is not clear that the predicted voxels are the outputs of the encoding procedure (they either look like the video models’ input or at best as if they are their direct output which is also not the case). (4) Finally, a major point of this figure was to show that different degrees of connectivity between regions are predicted, but from the size of the visualization on the left it is near impossible to tell differences between the thickness of the arrows.
* In section 3.4, where the Inter-intra Region Connectivity Model is introduced, there are no equations to show the model layers, and more importantly the loss function to train this module is not described at all, in equation or text form.
* In section 4.1, it is not clear whether the hyper-parameter tuning is conducted on the Algonauts training set or the authors’ training set (90% of the Algonauts training set, different for each of the 4 folds). This is an important clarification to make because in the first case the encoding test set is seen during hyper-parameter tuning.
* Phrases like “we mainly use the training set” in section 3.1 are too vague - when exactly do the authors not use it? From section 4.1 it seems that only the training test is used in all cases.
* Video models used are described as “video understanding” models throughout the paper, which is inaccurate because they are all video action recognition models - video understanding is a vague umbrella task that could potentially include segmentation, reasoning, and many other more complex tasks than classification. By also describing the image models as “image understanding” models, authors obscure the fact that they are trained on a different task than the video models, which is object recognition rather than action recognition.

References:

Conwell, Colin, et al. "What can 1.8 billion regressions tell us about the pressures shaping high-level visual representation in brains and machines?." BioRxiv (2022): 2022-03.

Conwell, Colin, et al. "A large-scale examination of inductive biases shaping high-level visual representation in brains and machines." Nature Communications 15.1 (2024): 9383.

**Questions:**

Overall the reviewer feels that although some of the presentation problems could be corrected, there are overarching problems with the structure, i.e. disjoint two parts, and the soundness of this paper, i.e. correlational evidence, unclean comparisons, and rushed conclusions, that make it hard to change in a way sufficient for acceptance.
The final rating is 5 in absence of the option for 4 (because 3 would be too low).

---

> ### Author Response · Authors · 2024-11-24
> **Response to Weaknesses - Part 1**
>
> **Weaknesses:**
>
> **1- “ paper reads as disjoint parts”.** Our two contributions are related and serve the same objective. They include both the first large-scale study of deep video understanding models on video stimuli data in addition to a novel method that is based on inter- and intra-region connectivity priors (L81-93). There exists an interplay between dynamics modeling and connectivity priors. First, connectivity priors show benefits on average in most of the models and regions when video understanding models are used more than image understanding ones, Fig. 6b  and Appendix B6 (Fig. 16). Second, both contributions provide better understanding of the visual cortex, deep learning models, and their intersection. It is insufficient to study the human brain without looking into both connectivity and dynamics which was also confirmed in our results, Fig. 3a, 5a and 15  and L365-369 and L426-429, and insights, L464-469. Figure 1 was edited to clarify the interplay as it shows our proposed approach adds the connectivity priors on the top of the video understanding models.
>
> **2- “Improvements in brain predictivity are caused by … connectivity, and not just less noisy embedding by considering other regions.”** provided in the common response.
>
> **3- Section 4.5 and Fig.6b only two models.** We use 3 models (MViT, SlowFast, and ResNet50) in our connectivity experiments because we focus on showing the performance enhancement with our best models (MViT and SlowFast), while showing a baseline model (ResNet50). Nonetheless, we add connectivity results with eight additional models in Appendix B6 (Fig. 15), still confirming our findings with statistical significance that connectivity improves neural encoding. We also include SlowFast in the comparison with respect to performance gain which confirms our findings in Appendix Fig. 16a.
>
> **4- Use of Algonauts challenge for the main results.** We use the training set since the test set was not provided as part of the challenge’s public release. Additionally, the Bold Moments Dataset was released concurrent to our work (July 2024) Lahner et al. (2024). Accordingly, we use the Algonauts data for the main work, but we also confirm our results on BMD, Appendix B1. Also the Algonauts data consists only of 9 regions which is easier to grasp and analyze in terms of explainability, than 22 regions (BMD Ver.A) or 47 regions (BMD Ver.B). Nonetheless, we add more results on BMD for connectivity results on VersionA which consists of 22 regions that are close to the ones used in the Algonatus dataset, Appendix Fig. 8.
>
> **5- “The supervised vs. self-supervised comparison is not clean.”** Fig. 3b shows a comparison between the same model pre-trained with self-supervision and fine-tuned with full supervision. It confirms our finding that fully supervised methods are on average better than self-supervised ones across all regions. We leave it for future work to explore more convolutional self-supervised models. Our results are already clean and the comparison between groups/families has been established previously by Han et. al. 2023 (ICML) but on image understanding.  Even when the models inside one family have some commonalities and differences. We also quote reviewer **LsQK: “The paper is of good quality, with good statistical analysis and control conditions.”**
>
> **“The paper results are shown with models mixed from different training datasets.”** This is inherent from the fact that image understanding (e.g, object recognition) and video understanding (e.g., action recognition) are trained with different datasets. Nonetheless, to reduce this impact we use multiple datasets in the video understanding (Kinetics, SSV2, Charades) to ensure the results are not just biased towards one dataset favoring video understanding models. We also provide statistical significance results when comparing between groups (families) of models that have similarities in terms of architecture or dynamics encoding but differences for example with respect to the training dataset. We use the tailored Welch’s t-test method to such a setup, since it is an unpaired test that works when sample sizes across groups are unequal and might have unequal variances.

---

> ### Author Response · Authors · 2024-11-24
> **Response to Weaknesses - Part 2**
>
> **6- Figure 4 is too fine-grained.** Being fine-grained does not go against being scientifically valid given they are under the same control setting (e.g., backbone, dataset, training and so forth). Also two-stream architectures and multiscale ViTs are critical design choices given their importance in the literature of video understanding models.
>
> **Two-stream vs. single-stream.** The two-stream architecture is a biologically-inspired architecture. Hence, it is important to examine the effect of the two-stream pathway on neural predictivity. **TimesFormer choice.** We mainly focus on comparing OmniMAE pre-trained (self supervision) vs. fine-tuned (full supervision), yet we add TimesFormer as another fully supervised video understanding model trained on the same dataset for wider context. **MViT results.** For the multiscale component in MViT, we hypothesize that it is an important contributor to the higher neural predictivity in early regions given that it is the main component that differentiates MViT from the other models in the transformer family. This hypothesis is also supported by our findings in the simulated environment as shown in Fig. 9a in Appendix B2. Fig. 9a shows a high similarity between MViT and convolutional models than ViTs. **Convolutional vs. transformer models results.** Our study is focused on presenting a statistically robust large-scale comparison between video understanding models and the human brain. We provide our novel findings, insights and possible explanations. The text L372-396 provides these as probable explanations to be further studied in future work. However, the finding that convolutional models are better than transformers in early regions has strong evidence with statistical significance already.
>
> **7- Figure 1.** We edited Figure 1 to address these comments
>
> **8- Connectivity equations.** We use basic fully-connected and dropout layers in the connectivity model in addition to a standard regularized regression loss. We described in the manuscript the layers used in L292-294 and updated the text with a description of the loss, L295.
>
> **9- Hyper-parameter tuning.** It is conducted with two-fold cross-validation on 90% of the Algonauts training set (different at each fold). The testing set was not seen during hyperparameter tuning, L315-316.
>
> **10- Vague sentences.** This phrase is used to clearly state that the training set of each dataset (1000 videos) is used in their respective experiments with four-fold cross-validation strategy, corrected in L211-213. We further clarify in Sec. 4.1 that we divide this set into two 90%-10% for training and testing.
>
> **11- “video understanding” vs. “action recognition”.** Any action recognition model is a video understanding model so the description of a “video understanding” model is accurate. Similarly, for object recognition and image understanding. Both are trained on classification tasks, but video understanding models take clips as input in order to learn dynamics unlike image understanding models that take static images as input.

---

> ### Author Response · Authors · 2024-11-24
> **Response to Questions**
>
> **Questions:**
>
> **“Overarching problems with the structure, i.e. disjoint two parts, and the soundness of this paper, i.e. correlational evidence, unclean comparisons, and rushed conclusions.”** Our work is coherent and studies the interplay of dynamics based neural encoding and connectivity priors with unified goals. Our comparisons are valid and follow previous literature while providing large-scale comparison (37 models), different target models as artificial neural networks and the human visual cortex. We provide, quoting **reviewer DZk6, unprecedented scale** while providing the full results with statistical significance, error bar plots and proper hyper-parameter tuning. We hope that the reviewer will value these contributions to the neuroscience literature when such scale, breadth and depth with focus on video stimuli and video understanding is missing in such cutting edge direction. Additionally, we provide multiple levels of analysis across groups of families and across variants of the same model (fine-grained analysis) to confirm our findings. Our conclusions are verified across both simulated and real experiments and in a study involving two million regression fits. In fact most of our conclusions are based only on statistically significant results. Hence, it confirms the soundness of our setup and experiments.

---

> > ### Comment · Area_Chair_mDze · 2024-11-25
> >
> > Dear Reviewer,
> >
> > The authors have provided their responses. Could you please review them and share your feedback?
> >
> > Thank you!

---

> > > ### Comment · Reviewer_MMrh · 2024-11-26
> > >
> > > Thank you for the response and clarifications. Although some of the issues were addressed, the original rating and opinion of this reviewer still stands due to fundamental structure and soundness factors.
> > > Concerning the structure of the paper, it still feels as two disjoint parts, and there was nothing done in the updated version to remedy that.
> > > As for the paper’s soundness, elaborating on each of the points made in the original review:
> > > * Thank you for providing additional figures in the appendix (Figures 17, 18), they are helpful in illustrating the method’s effect. Also for the clarification about LOC, indeed the directionality aspect of the matrix was not considered. However, an explanation of why LOC isn’t the source region for any target region is still lacking, as it is very noticeable in the matrix.
> > > * Thank you for adding more models in the appendix (Figures 15, 16). As for the connectivity improvement (figure 15), the additional results further validate the original conclusions of the authors. However, regarding the larger gain in the temporal models (figure 16), they validate this reviewer’s point that the conclusion “shows the necessity for encoding dynamics to fully benefit from such connectivity priors” cannot be inferred from testing only one image model. In all additional examples with one image and one video model (so 16a, 16d, and 16e) their difference is either not significant, or significant in favor of the image model (16e). Additionally, in the main text’s figure (6b) the improvement of the video transformer (MViT) is compared with that of an unrelated image CNN, while the most closely related image model would be the image ViT (16e), so the motivation behind this choice is unclear or even opportunistic. Finally, the comment about the improvement not being relative to models’ initial performance was not addressed.
> > > * The biggest issue here, apart from performing a comparison of this scale on a preliminary version of a recently released final dataset (BMD), is that even when using the BMD dataset (in the appendix at that), the results are still shown on the low-repetition training set instead of the test set developed with 3x the stimulus repetitions specifically for this purpose. The low repetitions of the training set significantly impact the SNR of the fMRI data, and are not suitable for evaluating models and making conclusions - this data is only suitable for training the encoding models.
> > > * Quoting another reviewer does not make the objections of this reviewer any less valid, especially objections based on concrete observations. These being, that you cannot safely draw conclusions based on a single model or by the unclean comparison of many models where other factors correlate with the factors examined (e.g. SSL with transformers). This should have been at least discussed in the limitations if it cannot be mitigated. Having said that, there is not even a limitations section on the main paper. About the training dataset, this reviewer’s point is that the main results should be controlled in a single dataset to be able to decouple the effects of dataset and modeling.
> > > * What fine-grained refers to in the review, was that conclusions are made again by comparing single models - a single two-stream architecture, a single multi-scale one, etc. Comparing TimeSformer with OmniMAE-fine-tuned, there are also other differences apart from the self-supervised pretraining (e.g. in the architecture) so there a concrete conclusion is also not possible.

---

> > > > ### Author Response · Authors · 2024-11-28
> > > > **Round 2 Response - Part III**
> > > >
> > > > **”There is not even a limitations section on the main paper.”** Section C details limitations, we can augment it with the inherent limitation that exists in all the literature comparing DNNs to biological neural systems in the camera ready.
> > > >
> > > > **“conclusions are made again by comparing single models”:** We have clarified earlier that this is a false claim. The majority of our conclusions are based on comparing families of models.
> > > > **“a single two-stream architecture”** We report two sets of models that are trained on Charades and Kinetics which show sufficient statistical significance using the paired t-test with exactly the same training data, optimization method, architecture and sampling rate. We leave it for future work to explore more two stream architectures and we plan to add this to the camera ready to be more accurate on this finding.
> > > > **“a single multi-scale one:”** our findings and conclusions were specific to MViT, we did not generalize to all multiscale models refer to Sec. 4.6.
> > > > **“OmniMAE-fine-tuned”** Again we first concluded the comparison between self supervised and fully supervised from the comparison among families of models. Then we confirmed it in a fine-grained analysis for OmniMAE trained with self supervision and full supervision.

---

> > > > > ### Comment · Area_Chair_mDze · 2024-11-30
> > > > >
> > > > > Dear Reviewer,
> > > > >
> > > > > The authors have provided their responses. Could you please review them and share your feedback?
> > > > >
> > > > > Thank you!

---

> > > > > > ### Comment · Reviewer_MMrh · 2024-12-02
> > > > > >
> > > > > > The authors’ response fails to address the issues raised by this reviewer. Specifically, the authors refuse to admit any fault or weakness of their study, even when the results point to the opposite of their conclusions, such as in figure 16e. Further, each of the reviewer’s points quoted is taken wildly out of context when answering, e.g. by referring to another analysis rather than the one that the reviewer’s point is about.
> > > > > >
> > > > > > Also, it is worth clarifying that:
> > > > > > - Statistical significance does not immediately make a result informative or publishable, it needs to be consistent across a sufficiently large and controlled set of samples.
> > > > > > - The notion of “clean” comparisons is, of course, a spectrum; it can almost never be perfect, due to hard-to-control model differences such as optimizers or embedding and tokenizing details. However, the factors left uncontrolled in this paper (architecture when comparing learning paradigms, and dataset when comparing dynamics modelling) are major and do not fall under this category.
> > > > > >
> > > > > >  Therefore, the original rating of this reviewer will not change.

---

> ### Author Response · Authors · 2024-11-28
> **Round 2 response - Part I**
>
> **“it still feels as two disjoint parts”** We have already clarified how our work is coherent based on two main reasons, the reviewer did not tackle our response and instead referred to feelings! We would appreciate a more concrete reason so we can provide a better response. We did not update the paper because we believe it is a coherent story already as provided earlier.
>
> **“Original rating and opinion of this reviewer still stands due to fundamental structure and soundness factors”** We argue that the reasons provided by the reviewer are not fundamental in nature and are quite fine-grained and do not affect the soundness of our work. Our summary of findings are only based on statistically significant results within unprecedented large-scale comparison. Looking into the majority of ICLR accepted works which had no statistical significance tests to begin with. ***If the ones basing their findings on statistically significant results with such depth and breadth of experiments are labelled not sound without providing clear evidence then this would be an unfair assessment! Leaving open questions that are not core to our study to future work exploration does not undermine our work or make it not sound either. Having an inherent limitation to our study that exists in all the previous well established studies on DNNs and its relation to biological neural systems yet again does not make our work not sound, but is simply a limitation that is remedied as detailed below.***
>
> **1-“Why LOC isn’t the source region for any target region is still lacking”** while this is a minor point that has no impact on the soundness of our work, we show here LOC column (as a source) in this connectivity matrix but when using only inter-region connectivity without intra-region; ***[V1:0.13, V2:0.14, V3:0.14, V4:0.18, LOC:0, EBA:0.39, FFA:0.35, STS:0.28, PPA:0.22]***. Without considering intra-region connectivity, LOC and other regions by design can’t contribute to themselves but LOC seem to contribute to mid-late regions at this point. Hence, we believe the reason in the full inter-intra region connectivity not showing LOC being source to other regions is simply being overwhelmed by other regions' contributions. This includes other regions’ contributions to themselves. These are learned connectivity not correlation based.
>
> **2-Fig.16 “Testing only one image model”.** We do provide both ViT and R50 results in pairs of comparisons as standard.
>
> **3-“The conclusion “shows the necessity for encoding dynamics to fully benefit from such connectivity priors” cannot be inferred”.** Here we directly compare the regression scores of SlowFast/MViT with connectivity that surpasses R50 with connectivity. Thus, this conclusion holds regardless.
>
> Showing as   V1,   	V2, 	V3,  	V4,   LOC,	EBA,  FFA,   STS,   PPA
>
> ***SlowFast: [0.301^,0.305^,0.288^, 0.285^, 0.311^,0.360^,0.293, 0.227^, 0.204]***
>
> ***MViT:  [0.297, 0.299, 0.283, 0.272, 0.316, 0.369, 0.306,	0.232, 0.218]***
>
> ***R50: [0.265, 0.265, 0.246,	0.249, 0.283, 0.317, 0.289, 0.202, 0.196]***
>
> (Stat. Sig. is highlighted with ^, indicating that SlowFast is  significantly surpassing R50 another convolutional model)
>
> Similarly, MViT w/ connectivity compared to ViT w/ connectivity with similar regions order:
>
> ***MViT Connectivity: [0.297^, 0.299^, 0.283^, 0.272^, 0.316^, 0.369^, 0.306^, 0.232^, 0.218^]***
>
> ***ViT Connectivity: [0.254, 0.241, 0.215, 0.217, 0.238, 0.268, 0.248, 0.181, 0.189]***
>
> Not only that, looking again into Fig. 16, this is our interpretation that dynamics encoding helps in full utilization of connectivity with at least 1-2 statistically significant regions being sufficient for this. It shows in Fig. 6b (4 regions w/ significance), 16a (2 regions  w/ significance), 16d ( 1 region w/ significance) the video model gain is more than image ones. However, we do an honest job in reporting that there is another interaction of how transformers generally gain more from connectivity (Fig. 16b, 16c, 16e). When this interaction is removed (Fig.16a, 16d) it still shows at least 1-2 stat. sig. results. On average most of the regions show better gains in video understanding models. Note that although MViT is a transformer based model it has been confirmed that it acts closer to convolutional models than transformers from real and simulated results, Fig. 3b and Fig.9a and our finding in summary (iv) with statistical significance. This explains the results of (Fig. 16e) that it stems from the difference in convolutional (or close to convolutional i.e., MViT) vs. pure transformers behaviour (i.e., ViTs). ***Being declared as opportunistic for the above when we report results across 3 video understanding models and 2 image ones is unfair assessment. Additionally, being rejected for this when we already show with strong statistical significance across two datasets that dynamics encoding is better and our connectivity is better, is unfair assessment too*** Fig. 16 results are only comparing the gain

---

> ### Author Response · Authors · 2024-11-28
> **Round 2 response - Part II**
>
> **“preliminary version of a recently released final dataset (BMD)”** We reported and confirmed our large-scale and connectivity results using both the Algonauts challenge dataset and the recently released BMD dataset. The fact that it was recently published is in our favour since it was concurrent work.
>
> **“Results are still shown on the low-repetition training set”**. We do report our results on BMD test set Ver.A (10 repetitions) comparing MViT w/ connectivity vs. w/o connectivity clearly validating our results with strong significance. Statistical analysis in the test set is conducted across subjects, since there are no folds in this case.
> Refer to updated manuscript Fig. 19.
>
> results are sorted according to the regions as: V1v, V1d, V2v, V2d, V3v, V3d, hV4, EBA, FFA, OFA, STS, LOC, PPA, RSC, TOS, V3ab, IPS0, IPS1-2-3, 7AL, BA2, PFt, PFop. Significance of connectivity surpassing without connectivity highlighted using ^.
>
> ***MViT w/o connectivity: [0.407, 0.447, 0.414, 0.450, 0.389, 0.421, 0.409, 0.516, 0.449, 0.475, 0.321, 0.468, 0.314, 0.218, 0.427, 0.392, 0.383, 0.313, 0.264, 0.222, 0.239, 0.205]***
>
> ***MViT w/ connectivity: [0.418^, 0.465^, 0.410, 0.471^, 0.397, 0.442^, 0.414, 0.550^, 0.494^, 0.513^, 0.354^, 0.503^, 0.353^, 0.258, 0.455, 0.427^, 0.424^, 0.370^, 0.314^, 0.278^, 0.288^, 0.257^]***
>
> **“This data is only suitable for training the encoding models.”** We refer to multiple well established works that conducted their study on 3 repetitions data; (Conwell et. al. 2024), which the reviewer referred to as clean and good study. We dont believe this is a reason for rejection and the reviewer is inflating it.
>
> *Conwell, Colin, et al. "A large-scale examination of inductive biases shaping high-level visual representation in brains and machines." Nature Communications 15.1 (2024): 9383.*
>
> **“You cannot safely draw conclusions based on a single model or by the unclean comparison of many models where other factors correlate with the factors examined (e.g. SSL with transformers). “**
>
> (1) We do not draw conclusions from one model referring to our summary of findings Section 4.6. pt. (i-iii) are based on comparison of families of models among 37 models, pt. (iv) is based on both simulated and real experiments comparing families of models among 37 models, pt. (v) compares two sets of models Slow vs. SlowFast trained on Charades and Kinetics, pt. (vi) is drawn from comparing a family of models then confirming it in the fine-grained experiment of OmniMAE w/ self supervision and full supervision, pt. (vii) we confirmed on 10 models. None of our main findings or conclusions is based on a single model!
>
> (2) The reviewer insists that comparing families of models is not “clean”, however referring to well established work in (Han et. al. ICML 2023) has conducted that procedure. ***By design any work comparing DNNs and its relation to neural encoding has this inherent limitation. We refer to Conwell et. al. 2024 comparing image understanding models which the reviewer labelled as clean. Yet by design they still had the following confounding factors in their comparison:***
> (i) optimization algorithms most of the recent transformers were trained with AdamW while old works in Convolutional approaches were mostly trained with SGD or Adam, even under same algorithm different hyperparameters that are tied to hyperparameter tuning differs. This entails a change in how the representations evolve over time beyond the architecture, which definitely impacts the final representations.
> (ii) another example include DeiT variants which was trained within a teacher-student training paradigm with access to teacher models that could be convolutional or a mixture of classifiers. Which can be viewed as having access to additional data (teacher data).
> (iii) The use of different input patchifying stem which consequently affects the contextual information captured from 16x16 to 32x32 across their studied transformer models.
>
> These are a few examples among others. These inherent limitations which affect all the literature conducting such large-scale studies are remedied by performing proper statistical analysis and by ensuring the breadth of the study. Both of which are covered in our work. In fact, we go the extra mile and report statistical significance while using a method that does not assume equal sized samples in the groups nor equal variances. ***Hence our comparisons are clean with the definition of what is clean from the reviewer themselves***
>
> (3) We remove multiple factors in our groups’ definition in the supplementary results in Fig. 11 and results still largely hold and are consistent, e.g., fully-supervised only in 11c.

---

### Official Review · Reviewer_LsQK · 2024-11-04

**Soundness:** 3
**Presentation:** 3
**Contribution:** 4
**Rating:** 6
**Confidence:** 3

**Summary:**

This paper compares a variety of image and video models to fMRI data, evaluating these models for representational alignment to human responses. The paper reports this analysis for multiple types of networks, comparing image/video, convolutional/transformer, single/multi stream, and self-supervised/unsupervised networks. The authors report the best predictors of human responses are convolutional, multi-stream, unsupervised video networks. Additionally, the paper reports experiments with a intra- and inter- region connectivity priors which improve encoding performance.

**Strengths:**

Originality: To my knowledge, this paper uses an established approach, but is novel in the scale and types of comparisons between types of models compared.

Quality: The paper is of good quality, with good statistical analysis and control conditions.

Clarity: The majority of the paper is written very clearly, especially the sections comparing image/video, convolutional/transformer, single/multi stream, and self-supervised/unsupervised networks to BOLD signal.

The scale of experiments in terms of number of models is commendable, and the authors were clever to choose models that enabled comparisons across many different dimensions.

Significance: This is an important emerging area in understanding alignment of models with human data, especially for new types of models, which show improved performance on visual tasks, but as the authors show, have worse representational alignment with human responses. This paper is significant in its contributions to understanding the quality of these alignments over multiple different axes of comparison.

**Weaknesses:**

There is little explanation of the intra- and inter- connectivity priors, especially their structure, and interpretation of their results beyond their contribution to improved representational alignment, and the value of their learned weights in figure 5c.

**Questions:**

I am a bit confused about the intra- inter- connectivity priors, what their structure is, and if the results demonstrate anything beyond that complex connectivity as seen in human visual cortex increases predictivity?

Relatedly, the paper would benefit from even a minimal interpretation of the learned connectivity weights. What is their structure and how does this change with training? How do they contribute to improvement of the representational alignment? What do these learned weights imply biologically? Such an analysis would likely also address the previous point.

I challenge the authors in L435-437 on leaving the reasons behind the better alignment seen in fully supervised networks to future work. This is a major result of the paper and I challenge the authors to at least make some hypotheses or have a few sentences of discussion as to why this could be the case. The validation of such a hypothesis can be left to future work.

Minor syntax issues:
L151 “: e?”.”
L363: “This difference decrease as”

---

> ### Author Response · Authors · 2024-11-24
> **Response**
>
> We thank the reviewer for their feedback.
>
> **Weaknesses:**
>
> **Connectivity related.**  please refer to the common response.
>
> **Questions:**
>
> **1- Connectivity related.  "What is their structure and how does this change with training?"** please refer to the common response, connectivity architecture and training, **"How do they contribute to improvement of the representational alignment? What do these learned weights imply biologically?"** Please refer to the common response, insights.
>
> **2- Fully supervised results interpretation.** Indeed these are interesting results that emerged from our study. In order to understand the reason behind this it requires studying the gradients from specialized tasks (e.g., object/action recognition) vs. more general tasks in self supervised learning (e.g., masked modeling) and how they affect the learned representations. It could be related to the specialization in the fully supervised tasks but it needs further exploration. Since our scope is focused more on the dynamics based encoding and connectivity prior, we leave this to future work.
>
> **3- Minor syntax issues.** We reflected this fix in the revised manuscript.

---

> > ### Comment · Area_Chair_mDze · 2024-11-25
> >
> > Dear Reviewer,
> >
> > The authors have provided their responses. Could you please review them and share your feedback?
> >
> > Thank you!

---

> ### Comment · Reviewer_LsQK · 2024-11-26
> **Response to Rebuttal**
>
> I appreciate the author’s responses and updates to the manuscript.
>
> 1) Connectivity related: Figure 1 was only very slightly updated (labels moved below rather than above boxes and arrows with varied thicknesses) The new version does little to improve clarity on the structural details of the iner-intra region connectivity model (layers, loss). While some information on training details are discussed in the common response, I still cannot find model details in the updated paper, nor do I see them addressed in the common response.
>
> 2) The goal of this feedback was not to have the authors perform additional experiments, but rather to encourage the authors to make hypotheses about why this MIGHT be the case, as part of the discussion. This would have strengthened the paper.
>
> In light of the other reviewer’s comments as well as the author’s rebuttals and paper updates, I believe there are many other outstanding issues that I was not aware of during my initial review, especially in lack of technical details. While I still believe this paper is above the acceptance threshold given the contributions, I believe my original rating was over-inflated and I change my score to a 6.

---

> > ### Author Response · Authors · 2024-11-28
> >
> > 1- We clarify in Fig. 1 the connectivity architecture which is simply as two fully connected layers with dropout with input from the predictions of 9 regions. We clarify the training procedure in the updated manuscript in text. Adding the training procedure in the Figure is not feasible.
> >
> > 2- We presented an initial hypothesis indeed as outlined in the above response which we can not confirm.
> >
> > We urge the reviewer to look into our responses for any outstanding issues from this review or the others' and we thank the reviewer for their positive feedback.

---

### Official Review · Reviewer_DZk6 · 2024-11-04

**Soundness:** 3
**Presentation:** 2
**Contribution:** 3
**Rating:** 6
**Confidence:** 3

**Summary:**

This work provides a comprehensive study of predicting fMRI signals in visual cortical areas while humans view short video clips, by mapping from the activations within a wide range of artificial neural networks that process the same videos or still images from them. The study compares models that differ along several interesting dimensions:  image-computable and video computable models, CNNs and transformers, and fully-supervised vs self-supervised training regimes. The results show that video-computable, convolution-based, fully-supervised models are generally the best predictors of visual cortical activity. The relevance of these results is supported by additional encoding studies where the encoding “target” is an artificial neural network of known architecture, showing that it is possible to distinguish between image-computable and video-computable models using a system identification approach. A novel strategy for incorporating inter and intra-region functional connectivity information is proposed, and this approach improves the encoding performance of the already best-performing models.

**Strengths:**

1.	Neural encoding is an important area of research that can yield insights to both improve artificial neural networks and better understand biological neural systems.

2.	There is a thorough review of recent related work that makes it clear how the current paper is situated given existing literature.

3.	A wide variety of appropriately chosen “source” candidate models are employed for comparative encoding experiments, and they vary in several interesting ways as outlined by the authors (single-image vs video computable, CNN vs transformer, fully supervised vs self-supervised). The breadth of the study in this regard is unprecedented for encoding of visual cortical responses to video inputs.

4.	The work demonstrates in a convincing way that video computable, fully supervised CNN models can generally enable more accurate predictions of visual cortical activity than models that are image computable, are based on transformer architectures, and/or are self-supervised.

5.	The work also presents evidence that system identification of image-computable vs video-computable systems is plausible (figure 2). This provides additional support for conclusions drawn regarding biological targets.

6.	A novel approach for incorporating inter and intra-region connectivity is presented, which significantly improves the voxel prediction performance.

**Weaknesses:**

This paper is primarily limited by a frequent dearth of clarity in the writing, as well as a methodological concern about the way inter and intra-region connectivity priors are incorporated into the encoding models. This reader’s score could improve considerably with clarification of these issues.

1.	Additional proofreading/editing is recommended to clarify and improve sentence structure in many parts of the paper. For example, the description of the “Real environment” (last part of section 3.1) is ambiguous – it states that two different datasets are used, but then says “we mainly use the training set, and perform cross-validation…”, “The dataset provides fMRI recordings of ten subjects…”, “each video and voxel in the brain was represented by a single activation value” without indicating which dataset these statements refer to (or is it both?). Also in this part of the paper, what does it mean that each video and voxel in the brain was represented by a single activation value? Aren’t there different values across time and between subjects? In several parts of the paper, there are similar ambiguities such as it being unclear which dataset is being referred to at what time.

2.	It is not clearly specified how the inter/intra region connectivity module is combined with the rest of the encoding models to make voxel activity predictions. The current version of the paper claims that intra-region connectivity priors improve predictive performance – but seemingly by directly providing the voxel values that are being predicted as inputs to the overall model. The improvements in performance might be due not to a meaningful incorporation of connectivity priors, but rather undesirable leakage of label information into the input.

3.	One of the claims is that inter/intra-region connectivity information improves performance to a greater extent in video-computable models compared with image-computable models – the evidence for this is presented in  figure 6b. However, there is a confound because connectivity-related performance improvements are compared between image-computable ResNet-50 and video-computable MViT: how can it be determined whether the apparent interaction effect is related to image computable vs video computable models rather than CNNs vs transformers?

Minor comments:

4.	In the abstract and several other places, saying that “the study encompasses more than two million regression fits” does not seem especially relevant – the number of regression fits in itself does not seem to be a particularly informative way to quantify the scale of the study.

5.	In describing the contributions of the paper, the authors seem to emphasize that prior works did not compare different neural network architectures - but in section 3.1, there is a statement that “while previous works focused on the architecture aspect, we argue it is even more important to look into whether the model is learning dynamics (e.g., motion) or simply using static information from a single image.” Yet, the comparison between single image vs video processing would seem to be an important emphasis of Lahner et al 2024. These claims about the relationship between prior works and the current paper should be clarified.

6.	The terms “real environment” and “simulated environment” are not especially informative – perhaps something like “biological target” and “artificial neural network target” would be an improvement

7.	In first paragraph of section 4.1, “As for image understanding models we use the default sampling rate of eight.” Does this refer to 8 frames per second? This whole paragraph seems quite ambiguous – why are video understanding vs image understanding models being discussed here in relation to sampling rates?

8.	The text is some of the figures is too small to be easily readable, especially figure 2, figure 3, figure 5, and supplementary figures 7, 8, 10, and 11.

9.	Most of the in-text citations would be best formatted with both author and year inside the parentheses. In LaTex, you can use \citep{} instead of \cite{}

10.	The goal of the paper seems to be stated in an overly general way in the abstract (“This comparative analysis aims to advance and interpret deep neural networks and enhance our understanding of biological neural systems”).

11.	Typos:

A.	In the introduction (end of page 1), “while the later showed that these fMRI recordings” (should be latter instead of later)

B.	Grammar issue in start of 2nd paragraph of section 3.1: “How do deep video understanding models families compare to biological neural systems”

C.	End of section 3.1 “The datasets we use is provided at TR one second” (should be are, not is, and the TR acronym should be defined)

**Questions:**

1.	The following reflects the evolution of my understanding while reading the paper: the authors study the ability of artificial neural network activations to predict biological signals in the visual cortex (the “target”) – they also include an ANN-based target – what does the artificial target add to the study? Section 3.1 seemingly attempts to explain this, but it may need to be edited for clarity. Is the idea that we know the structure of the target ANNs in the artificial setting, so we can test our ability to uncover mechanistic insights through encoding experiments, and therefore get a sense for how informative the system identification experiments with biological targets are likely to be? This becomes much clearer upon review of Han et al 2023, but it should also be explained more explicitly in the introductory sections of the present work.

2.	In equation 1, in the first term, is a 1/L normalization needed for the summation $\sum_{l=1}^{L} \omega_l \hat{Y}_l$ ? Without the normalization, it seems mildly inconsistent with the statement above equation 1 that “we learn the weights of one fully connected layer to provide the predictions of the voxels of one region of interest in the visual cortex as, $\hat{Y}_l = W_l X_l$.” Or perhaps this normalization is absorbed by $\omega_l$?

3.	In section 3.4, the description of the connectivity module indicates that all voxels from all regions are used to predict voxels from a single region. Doesn’t this mean that the desired output of the model (voxel values from a single target region) are contained in the input of the model, making this a trivial mapping? It makes sense that dropout layers could partially alleviate this problem, but why does it make sense to set up the architecture in this way in the first place where at least some of the outputs are trivially accessible in the input?

4.	Section 4.1: for four-fold cross validation where each fold has 90% of the videos in the training set and 10% in the test set – are the 10% test sets disjoint between the folds, or randomly selected? (Not a major issue, but 75:25 might have been a slightly more natural choice for four-fold cross validation)

5.	In section 4.2, there is a statement that “we include video understanding models that are trained on three different datasets which are Kinetics, Charades, and Something-Something v2”. These datasets are not cited, and it is not clear how the results from the three datasets were combined to produce figure 2 – were the results pooled somehow?

6.	Might it be useful to discuss why the observed trends seem to hold for some visual cortical regions and not others? For example, video-computable models are no better than image-computable models for predicting FFA and PPA activity - it is interesting to speculate that perhaps recognition of faces (associated with FFA) or environmental scenes (associated with PPA) in the brain might be less dependent on temporal dynamics.

7.  For plots like those in figures 2 and 3, what does each dot represent? For example, is it one regression score from one type of “source” model averaged among cross-validation folds, or is it an accuracy from one of the folds only? Are results from different source models pooled together in these plots? Relatedly, it is not clearly specified how the Welch’s t-tests are performed – which sets of values specifically are used to compute the t-test?

8.	Why are system identification results presented in the main text specifically for image vs video computable models, rather than other comparisons of interest like CNNs vs transformers or self-supervised vs fully-supervised?

---

> ### Author Response · Authors · 2024-11-24
> **Response to Weaknesses**
>
> **Weaknesses:**
>
> **1- “Additional proofreading/editing is recommended…”** These ambiguities have been addressed in the revised manuscript, L211-L215. We ran experiments per dataset, in each dataset we used the corresponding training set and performed cross-validation in 90%,10% train-test split over four folds randomly selected. Both datasets were provided in which the response to each video stimulus is presented as a single activation value (averaged/processed across time) per voxel per participant (Cichy et al., 2021, Lahner et al., 2024). Additionally, we applied temporal averaging on deep learning models’ representations for computational efficiency reasons. We ran regression per subject then provided averaged results over subjects per fold, followed by averaging over the four folds.
>
> **2- Clarifications on our connectivity model.** Provided in the common response.
>
> **3- Fig. 6b “How can it be determined…image vs video computable models rather than CNNs vs transformers?”** We thank the reviewer for their valuable comment and agree on the presence of confounding factors. To further explore the effect of this, we compare ResNet50 (image) vs. SlowFast (video) from the convolutional family in Fig 16a as requested. Similarly, we compare the connectivity model on ResNet-50 (image) vs. 3D ResNet-50 (video) and ViT (image) vs. MViT (video) from the transformers family, refer to Appendix B6 (Fig 16). Our results show that video based connectivity performance gain is on average higher than single image ones in the majority of the regions, with exception to early regions in ViT. Additionally, comparing convolutional with transformer models (ResNet50 vs ViT and SlowFast vs MViT) show the superiority of transformers in terms of connectivity performance gain, Fig. 16 b, e, especially in the early regions.
>
> **Minor Comments:**
>
> **4- “two million regression fits”** The number of regression fits has been previously used in the literature (Conwell, C., et al., (2021, 2022)). Regression fits reflect the study scale in terms of the number of source models, target models, regions and voxels.
>
> **5- Clarification on comparison to related works.** We clarified in L84-85 and L161-185 our relationship with other prior works that studied the time aspect and highlighted that our difference is being large-scale and comprehensive. We discussed in L84-85 that Lahner, et al., 2024 only studied 3 models with limited types while our work includes more than 35 models. Quoting the reviewer himself “The breadth of the study … is unprecedented” but indeed the statement referred to missed the emphasis on the large-scale and comprehensive aspect which was added in (L161-185).
>
> **6- “biological target” and “artificial neural network target” terms.** We changed these terms indeed as recommended.
>
> **7- Sampling rate clarification.** The sampling rate here does not refer to frames per second, it is rather the rate used to sample frames from the video clip. Following the video understanding literature, we follow this equation; number of frames in input clip = (video clip duration*frames per second)/sampling rate. Video clip duration and frames per second are parameters that are based on the video stimuli data. However, the sampling rate is a hyper-parameter tied to how the video understanding model was initially trained and for image understanding models we set it to eight. This detail is described in the manuscript in L310-312 and further clarifications added in Appendix A.
>
> **8-9 Formatting.** We changed these based on their recommendation in the revised manuscript, except for certain Figures which were not feasible to increase their font for space constraints, e.g., Appendix Fig. 7.
>
> **10- Abstract too general.** The sentence referred to in the abstract is not meant to describe our study, but describing the literature. The statements we use in the abstract, starting from L17, reflect what we show and prove in the manuscript in our results and findings.
>
> **11- Typos.** We fixed all typos in the revised manuscript.

---

> ### Author Response · Authors · 2024-11-24
> **Response to Questions**
>
> **Questions:**
>
> **1- What does the artificial target add to the study?** We explain in the paper the importance of adding the artificial target to study in (L151-153) and (L158-186).
>
> **2- Normalization in Eq. 1.** The 1/L normalization is a constant value that should not affect the backpropagation and learned weights.
>
> **3- Connectivity vs. trivial mapping.** Our connectivity is not a trivial mapping since we do ablate intra connectivity (within region voxels) vs. inter connectivity (across regions) vs. full connectivity (inter and intra-region) in Fig 5b. It confirms our full connectivity model surpasses both as it utilizes other regions not only trivially mapping input (same region voxels) to output. Second, we present in Table 11 low regression scores when using identity connectivity weights. Finally, as we mentioned in L302-303, the ground-truth activations were only used in the training phase on training data, not during inference which is evaluated on separate test data per fold.
>
> **4- Are test sets disjoint between the folds, or randomly selected?** The 90%-10% split is randomly selected per fold, but the test set is disjoint from the training set per fold (i.e. no leakage between training and testing per fold). We followed Zhou et al. (2022) in the 90%-10% training-test splits.
>
> **5- Fig. 2, datasets results.** We have citations of the datasets in Table 1 which encompasses all the models we use throughout all the experiments except the fine-grained ones which were selective. The results of these datasets are included in the comparisons of families of models (each represented separately). We include models trained on different datasets to reduce the effect of the dataset, as a confounding factor in our results.
>
> **6- Differences across regions.**  Yes, we agree with the reviewer that these results might be caused by the less dependence of some regions (such as FFA and PPA) on temporal dynamics. This is also supported by previous literature which shows that FFA computes static properties of faces, unlike STS which is more responsive to dynamic information in faces (Pitcher, D, 2011). The PPA was also found to show no difference in responding to static vs dynamic stimuli, unlike EBA which was found to be more responsive to dynamic stimuli than static stimuli (Pitcher, D, 2019). Our results are consistent with these previous findings where video models are better than image ones in STS and EBA with significance while in FFA and PPA, the results are not statistically significant. We updated the manuscript with this insight L367-370.
>
> *Pitcher, David, et al. "Differential selectivity for dynamic versus static information in face-selective cortical regions." Neuroimage 56.4 (2011): 2356-2363.‏*
>
> *Pitcher, David, Geena Ianni, and Leslie G. Ungerleider. "A functional dissociation of face-, body-and scene-selective brain areas based on their response to moving and static stimuli." Scientific reports 9.1 (2019): 8242.‏*
>
> **7- Fig. 2 & 3 question.** Each dot represents the regression score of one source model regressing on one target averaged across the four cross-validation folds. The t-test in the figures is used to compare the scores of models (represented by dot for each) across the two groups.
>
> **8- System identification in main text.** We include the comparison of image vs video computable models in the main text because the main focus of this paper is comparing video-understanding and image understanding models. However, we include CNNs vs transformers comparison in Appendix Figure 9 and 10 for space constraints.

---

> > ### Comment · Area_Chair_mDze · 2024-11-25
> >
> > Dear Reviewer,
> >
> > The authors have provided their responses. Could you please review them and share your feedback?
> >
> > Thank you!

---

> > > ### Comment · Reviewer_DZk6 · 2024-11-30
> > > **Response to authors' rebuttal**
> > >
> > > My thanks to the authors for their responses. Some of my concerns have been addressed, notably about the value of the artificial target and a clarification about how the inter/intra region connectivity priors work (they use predicted voxel activations as inputs during inference, not the actual measured activations). However, I am not convinced by the response to weakness #3 from my original review. To recap, here's what I wrote:
> > >
> > > 3. _One of the claims is that inter/intra-region connectivity information improves performance to a greater extent in video-computable models compared with image-computable models – the evidence for this is presented in figure 6b. However, there is a confound because connectivity-related performance improvements are compared between image-computable ResNet-50 and video-computable MViT: how can it be determined whether the apparent interaction effect is related to image computable vs video computable models rather than CNNs vs transformers?_
> > >
> > > Here is the authors' response:
> > >
> > > _We thank the reviewer for their valuable comment and agree on the presence of confounding factors. To further explore the effect of this, we compare ResNet50 (image) vs. SlowFast (video) from the convolutional family in Fig 16a as requested. Similarly, we compare the connectivity model on ResNet-50 (image) vs. 3D ResNet-50 (video) and ViT (image) vs. MViT (video) from the transformers family, refer to Appendix B6 (Fig 16). Our results show that video based connectivity performance gain is on average higher than single image ones in the majority of the regions, with exception to early regions in ViT. Additionally, comparing convolutional with transformer models (ResNet50 vs ViT and SlowFast vs MViT) show the superiority of transformers in terms of connectivity performance gain, Fig. 16 b, e, especially in the early regions._
> > >
> > > Examining Fig 16a, I agree that there is support for the superiority of transformers over convolutional architectures in terms of connectivity prior-related performance gains. However, there is only very weak support for the notion that connectivity priors improve predictivity to a greater extent than for video-computable models than image-computable models: only 2 of the 9 differences are statistically significant - and are these tests corrected for multiple comparisons? Meanwhile, Fig 16e strongly argues against this conclusion. **Therefore, I conclude that this is _not_ a claim that can be made using the provided evidence.** This seems to be one of the central claims of the paper, stated in the abstract ("it also shows the necessity for encoding dynamics to fully benefit from such connectivity priors."), the main contributions list in the introduction ("We also show that encoding dynamics is an important aspect to enable the full utilization of such connectivity priors") and the finding is stated again at the end of Section 4.5. I recognize that this is not the only interesting finding from the paper. Given my current understanding, it is not clear to me that the paper can be published with these claims as stated, and it seems to me that they should be removed. I would appreciate any clarifications from the authors.

---

> ### Author Response · Authors · 2024-12-01
> **Round 2 Response**
>
> We thank the reviewer for their response. Below we provide further clarifications as requested by the reviewer.
>
> The results we present in the paper show that combining dynamics with connectivity priors is important for better neural predictivity performance, and also for a better understanding of the visual system. This is what we mean by “the necessity for encoding dynamics to fully benefit from such connectivity priors” and “encoding dynamics is an important aspect to enable the full utilization of such connectivity priors”.  This finding is clarified by directly comparing the regression scores of SlowFast with connectivity that surpasses ResNet50 with connectivity:
>
> Showing as V1, V2, V3, V4, LOC, EBA, FFA, STS, PPA
>
> ***SlowFast: [0.301^,0.305^,0.288^, 0.285^, 0.311^,0.360^,0.293, 0.227^, 0.204]***
>
> ***ResNet50: [0.265, 0.265, 0.246, 0.249, 0.283, 0.317, 0.289, 0.202, 0.196]***
>
> (Stat. Sig. is highlighted with ^, indicating that SlowFast is significantly surpassing ResNet50 another convolutional model)
>
> Similarly, MViT w/ connectivity compared to ViT w/ connectivity with similar regions order:
>
> ***MViT Connectivity: [0.297^, 0.299^, 0.283^, 0.272^, 0.316^, 0.369^, 0.306^, 0.232^, 0.218^]***
>
> ***ViT Connectivity: [0.254, 0.241, 0.215, 0.217, 0.238, 0.268, 0.248, 0.181, 0.189]***
>
> (We can add these comparisons in the appendix in the camera-ready version).
>
> In other words, although image models (such as ResNet50 or ViT) performance increases after incorporating the connectivity priors, they did not fully utilize the connectivity priors as they do not surpass the performance achieved by video models with connectivity priors (such as SlowFast or MViT).
>
> This is the main reason behind our claim as supported by our results. Additionally, in Figure 16, in terms of performance gain, the results show that on average video models have higher performance gain compared to image models. In Figure 16a, 6 out of 9 regions on average have higher performance gain (statistically significant in 2 regions) in SlowFast compared to ResNet50. In Figure 16d, 7 out of 9 regions have higher performance gain (statistically significant in 1 region) in 3D ResNet50 compared to ResNet50. Even in Figure 16e, 5 out of 9 regions on average have higher performance gain (statistically significant in 1 region) in MViT compared to ViT. However, we agree with the reviewer that Figure 16e is a bit different than Figure 16a and Figure 16d in the higher (statistically significant) performance gains in 4 early regions in ViT comparing MViT.  We think this could be affected by our finding that despite MViT is a transformer-based model, it has been confirmed that it acts closer to convolutional models than transformers from real and simulated results (with statistical significance), Fig. 3b and Fig.9a and our finding in, section 4.6, summary (iv). This could explain the results of (Fig. 16e) that it stems from the difference in convolutional (or close to convolutional i.e., MViT) vs. pure transformers behaviour (i.e., ViTs). We use paired t-test when comparing pairs of models and Welsh’s t-test when comparing groups of models.
>
> Accordingly, we believe that our claim about the necessity of combining dynamics with connectivity priors is valid, but we agree with the reviewer that we can better rephrase this claim. We can rewrite in the camera-ready version this finding/claim and the last paragraph of section 4.5 to better clarify that: 1) to benefit from connectivity priors, it is better to combine them with video understanding models as they achieve statistically significant higher scores than image understanding ones with connectivity priors, 2) video understanding models achieve higher performance gain on average in most of the regions, yet there is an interaction from the architecture type, explained in Appendix B.6.

---

> > ### Comment · Reviewer_DZk6 · 2024-12-02
> > **Follow-up reponse to authors**
> >
> > I thank the authors for responding to my concerns. I think an appropriate conclusion based on these results is that connectivity priors have a larger benefit (relative to image-computable models) for video-computable **convolutional** models, not in transformers/across architectures, and only for certain brain regions (V1 and PPA, according to Figure 16a). Additionally, in Figure 16e, MViT has a greater improvement from incorporating connectivity than ViT does specifically for the PPA brain region (but lesser improvement for V1-V4). Overall, the argument holds for the PPA brain region only (or for V1 and PPA in the case of convolutional models).
> >
> > Given the evidence presented, I do not think it is possible to make a more general determination than this about the relative benefits of connectivity priors for image-computable vs video-computable models. In the response above, the authors provide results comparing the absolute regression scores of ResNet50 (convolutional, image-computable) with SlowFast (convolutional, video-computable), showing that SlowFast outperforms ResNet50 with statistical significance across brain regions. However, this can **not** be directly used to argue that connectivity priors are somehow synergistic/of greater benefit for video-computable models - it could be that the performance difference is simply between image-computable and video-computable models, with video-computable models being superior in this context (which is one of the main points of the paper). The same reasoning applies to the comparison in the authors' response between ViT (transformer, image-computable) and MViT (transformer, video-computable) - moreover, Figure 16e argues strongly against the idea that connectivity priors help video-computable models more than image-computable models in general.
> >
> > In their response, the authors attempt to support a broader conclusion with additional statistical arguments, writing: ``In Figure 16a, 6 out of 9 regions on average have higher performance gain (statistically significant in 2 regions) in SlowFast compared to ResNet50. In Figure 16d, 7 out of 9 regions have higher performance gain (statistically significant in 1 region) in 3D ResNet50 compared to ResNet50. Even in Figure 16e, 5 out of 9 regions on average have higher performance gain (statistically significant in 1 region) in MViT compared to ViT.'' This statement is a very weak argument for connectivity priors having greater benefit in video-computable than in image-computable models - it is based on interpretation of non-significant differences - the statistical tests are supposed to determine which differences can be interpreted. It is also important to note that there appears to be no correction for multiple comparisons in this analysis, resulting in potentially inflated type I error rates given the large number of comparisons involved. I understand that it is too late to add experiments, but as a suggestion for a potential camera-ready version: one possibility to better support this kind of conclusion would be to do a joint analysis across brain regions using (for example) a linear mixed model with random effect intercepts for each brain region.
> >
> > The authors have made other clarifications that have improved the paper, and to reflect this I am willing to increase my score. To be clear, however, this is contingent on the authors agreeing to make a more limited version of the argument about ``the necessity for encoding dynamics to fully benefit from such connectivity priors.'' Unless I am seriously misunderstanding something, the current evidence cannot be interpreted to make this claim in a general sense.

---

> ### Author Response · Authors · 2024-12-03
> **Round 3 Response**
>
> We thank the reviewer for their response. We believe our results are on the side of the interaction between dynamics encoding and connectivity priors to an extent, but we can indeed limit it as the following: ***dynamics encoding positively interacts and benefits connectivity priors to an extent. Yet there is another interaction relating to the model's architecture type and brain region(s). Nonetheless, across various models, we show that combining dynamics encoding with connectivity priors surpass their counter-parts that rely on static information from single images even if equipped with connectivity.***
>
> We thank the reviewer for providing us with the chance to increase their score.

---

### Author Response · Authors · 2024-11-24
**Common Response**

We thank the reviewers for their constructive and insightful feedback and refer to some of their positive comments commending:

**(1) Our large-scale study.** DZk6: “The breadth of the study in this regard is unprecedented…to video inputs.” LsQK: “the authors were clever to choose models that enabled comparisons across many different dimensions.” MMrh: “In general this work is a step in the right direction.”
**(2) Our connectivity model.** DZk6: “A novel approach for incorporating inter and intra-region connectivity is presented”
**(3) Others.** DZk6: “There is a thorough review of recent related work.” LsQK: “The paper is of good quality, with good statistical analysis and control conditions.” MMrh: “the writing style has good quality.“

A revised manuscript with all edits highlighted in red is uploaded. Three reviewers had questions and concerns regarding the connectivity model which are addressed here:

**1- Clarifications on connectivity model (DZk6, LsQK) -** ***Architecture:***. The inter/intra region connectivity priors model is explained in the manuscript L291-294 and shown in Figure 1 which was edited for further clarification as requested from the reviewers.
***Training: (Stage 1)*** We train regression without connectivity where the video stimulus is used as input to the neural network from which intermediate representations are extracted and used for predicting voxel activations. ***(Stage 2)*** The inter/intra region connectivity module is then applied on the top by using the voxels of the nine regions as inputs to learn their connectivities. The ground-truth activations are used only in the training phase for learning connectivity weights while predicted voxels from all regions are used in the inference phase. Our connectivity training is elaborated in the revised manuscript L297-304.
***Insights:*** We have shown the importance of this connectivity model in improving regression scores (Figure 5a, b, and Figure 6) in addition to the biological explainability in terms of connectivity weights as shown in Figure 5c. In the manuscript, we clarify, in L464-470, the biological understanding that we can conclude from Figure 5c, in connection (Gilbert & Li, 2013). To further improve the biological interpretation of the model, we added the functional connectivity that is based on the correlation (not learned) between the regions, Appendix B6 (Fig. 17). Our learnable connectivity map provides information about the bidirectional relationship between the regions that is not provided by the correlation-based connectivity. Additionally, our findings are aligned with (Zeng, H., et al., 2023) showing: 1) the bidirectional information flow, feedforward, and feedback pathways in the visual cortex. 2) strong reciprocal connections between V1, V2, V3, and V4 regions.
***Notes:*** There is no leakage of label information in the connectivity model. First, as explained in L323,339, the data is divided into distinct training (90%) and testing (10%) parts with no data leakage between the training and testing splits per fold. Second, only the predicted activations from all regions are used as input during inference on the unseen 10% test set. Third, we present in Figure 5c the learnable connectivity weights which show the meaningful incorporation of connectivity priors as learned by the model. Fourth, we show in Table 11 the low accuracies achieved when using identity connectivity (simply a weight of one to all voxels in all regions) which proves that the connectivity model learned meaningful connectivity priors.

*Zeng, H., Chen, S., Fink, G. R., & Weidner, R. (2023). Information exchange between cortical areas: the visual system as a model. The neuroscientist, 29(3), 370-384.‏*

**2- “Improvements in brain predictivity are caused by…” (MMrh)** We support our finding that the performance improvement is caused by the learned connectivity in two ways: 1) We provide the learned connectivity matrix that shows the bidirectional connectivity across the regions, Fig. 5c. We compare this matrix with the correlation based (not learned) functional connectivity, in the updated manuscript, Fig. 17. While there is a similarity between the learned and correlation based connectivity maps, the learned connectivity provides further information about the bidirectional relationship between the regions that is not captured by correlations. 2) Table 11 shows low results when using identity connectivity weights, i.e., using weights as one. If the improvement was caused by considering other regions, the identity connectivity should have improved. **“LOC does not connect with any other areas”. (MMrh)** Figure 5c, the y-axis shows the “target regions” while the x-axis shows the “source”. Each row shows the contributions of the source regions in predicting the target. Accordingly, the matrix shows the mid and late regions are contributing to the predictions of LOC, hence why it improved.

---

### Meta-Review · Area_Chair_mDze · 2024-12-14

**Metareview:**

The paper presents a comprehensive study comparing image and video understanding models to visual cortex recordings using video stimuli. The study reveals insights into how different specs of deep networks predict neural responses. Moreover, the paper introduces a novel neural encoding scheme that leverages connectivity priors for improved performance.

The paper is easy to follow. The experiments benchmarking so many AI models and leveraging connectivity prior are well-motivated, original, and interesting.

The paper has mixed reviews with two 6s, 5, and 1.

After reviewing the rebuttal and initiating intensive internal discussions with all the reviewers, the Area Chair (AC) and reviewers collectively reached a consensus on the following weaknesses of the paper, which ultimately led to its rejection:

[Inconsistencies in results, not well-controlled experiments, and results fail to support the claims] The experiments comparing models are not well-controlled, and the results fail to support the claims made in the paper, such as those in Fig. 16e. Even after the rebuttal, the paper continues to overstate the positive effect of connectivity priors, without acknowledging that this effect is observed only in certain brain regions, not universally. While the authors have made minor adjustments to qualify their claims in response to other reviewers' points, they remain reluctant to fully address the nuances of their findings.

[Missing experimental details] The paper lacks clear details regarding the models, training procedures, and the specifics of the connectivity priors. Despite reviewers requesting these clarifications, the revised version does not include the necessary information.

[Inadequate responses to concerns during rebuttal] Based on the points above, it is evident that the authors largely dismissed these well-founded critiques. All reviewers agree that the authors persistently refused to acknowledge or adequately address these issues, either in their responses or in the manuscript itself.

**Additional Comments On Reviewer Discussion:**

The paper has mixed reviews with two 6s, 5, and 1.

After reviewing the rebuttal and initiating internal discussions with all the reviewers, the Area Chair (AC) and reviewers collectively reached a consensus on the following weaknesses of the paper, which ultimately led to its rejection:

[Inconsistencies in results] The experiments comparing models are not well-controlled, and the results fail to support the claims made in the paper, such as those in Fig. 16e. Even after the rebuttal, the paper continues to overstate the positive effect of connectivity priors, without acknowledging that this effect is observed only in certain brain regions, not universally. While the authors have made minor adjustments to qualify their claims in response to other reviewers' points, they remain reluctant to fully address the nuances of their findings.

[Missing experimental details] The paper lacks clear details regarding the models, training procedures, and the specifics of the connectivity priors. Despite reviewers requesting these clarifications, the revised version does not include the necessary information.

---

### Decision · Program_Chairs · 2025-01-22

Reject